# AdaMatch: A Unified Approach to Semi-Supervised Learning and Domain Adaptation

**David Berthelot**[†*]**, Rebecca Roelofs**[†*]**, Kihyuk Sohn**[†]**, Nicholas Carlini**[†]**, Alex Kurakin**[†]

[†] Google Research

## Abstract

We extend semi-supervised learning to the problem of domain adaptation to learn significantly higher-accuracy models that train on one data distribution and test on a different one. With the goal of generality, we introduce AdaMatch, a unified solution for unsupervised domain adaptation (UDA), semi-supervised learning (SSL), and semi-supervised domain adaptation (SSDA). In an extensive experimental study, we compare its behavior with respective state-of-the-art techniques from SSL, SSDA, and UDA and find that AdaMatch either matches or significantly exceeds the state-of-the-art in each case using the same hyper-parameters regardless of the dataset or task. For example, AdaMatch nearly doubles the accuracy compared to that of the prior state-of-the-art on the UDA task for DomainNet and even exceeds the accuracy of the prior state-of-the-art obtained with pre-training by 6.4% when AdaMatch is trained completely from scratch. Furthermore, by providing AdaMatch with just one labeled example per class from the target domain (i.e., the SSDA setting), we increase the target accuracy by an additional 6.1%, and with 5 labeled examples, by 13.6%.[1]

## 1 Introduction

Since the inception of domain adaptation and knowledge transfer, researchers have been well aware of various configurations of labeled or unlabeled data and assumptions on domain shift (Csurka, 2017). Unsupervised domain adaptation (UDA), semi-supervised learning (SSL), and semi-supervised domain adaptation (SSDA) all use different configurations of labeled and unlabeled data, with the major distinction being that, unlike SSL, UDA and SSDA assume a domain shift between the labeled and unlabeled data (see Table 1). However, currently the fields of SSL and UDA/SSDA are fragmented: different techniques are developed in isolation for each setting, and there are only a handful of algorithms that are evaluated on both (French et al., 2018).

Techniques that leverage unlabeled data are of utmost importance in practical applications of machine learning because labeling data is expensive. It is also the case that in practice the available unlabeled data will have a distribution shift. Addressing this distribution shift is necessary because neural networks are not robust (Recht et al., 2019a; Biggio & Roli, 2018; Szegedy et al., 2013; Hendrycks & Dietterich, 2019; Azulay & Weiss, 2018; Shankar et al., 2019; Gu et al., 2019; Taori et al., 2020) to even slight differences between the training distribution and test distribution. Although there are techniques to improve out-of-distribution robustness assuming no access to unlabeled data from the target domain (Hendrycks et al., 2020; Zhang, 2019; Engstrom et al., 2019; Geirhos et al., 2018; Yang et al., 2019; Zhang et al., 2019), it is common in practice to have access to unlabeled data in a shifted domain (i.e. the UDA or SSDA setting), and leveraging this unlabeled data allows for much higher accuracy. Moreover, while SSDA (Donahue et al., 2013; Yao et al., 2015; Ao et al., 2017; Saito et al., 2019) has received less attention than both SSL and UDA, we believe it describes a realistic scenario in practice and should be equally considered.

In this work, we introduce AdaMatch, a unified solution designed to solve the tasks of UDA, SSL, and SSDA *using the same set of hyperparameters regardless of the dataset or task*. AdaMatch extends FixMatch (Sohn et al., 2020) by (1) addressing the distribution shift between source and target domains present in the batch norm statistics, (2) adjusting the pseudo-label confidence threshold on-the-fly, and (3) using a modified version of distribution alignment from Berthelot et al. (2020).

---

[*]equal contribution
[1]Code to reproduce results: `https://github.com/google-research/adamatch`

| Task | Labeled | Unlabeled | Distributions |
|------|---------|-----------|---------------|
| SSL | source | target | source $=$ target |
| UDA | source | target | source $\neq$ target |
| SSDA | source+target | target | source $\neq$ target |

Table 1: Relations between the settings of Semi-Supervised Learning (SSL), Unsupervised Domain Adaptation (UDA), and Semi-Supervised Domain Adaptation (SSDA).

AdaMatch sets a new state-of-the-art accuracy of 28.7% for UDA without pre-training and 33.4% with pre-training on DomainNet, an increase of 11.1% when compared on the same code base. With just one label per class on the target dataset, AdaMatch is more data efficient than other method, achieving a gain of 6.1% over UDA and 13.6% with 5 labels. We additionally promote democratic research by reporting results on a smaller $64 \times 64$ DomainNet. This results in a minimal drop in accuracy compared to the full resolution, and compared to the practice of sub-selecting dataset pairs, does not bias the results towards easier or harder datasets. Finally, we perform an extensive ablation analysis to understand the importance of each improvement and modification that distinguishes AdaMatch from prior semi-supervised learning methods.

## 2 RELATED WORK

**Unsupervised Domain Adaptation (UDA).** UDA studies the performance of models trained on a labeled source domain and an unlabeled target domain with the goal of obtaining high accuracy on the target domain. Inspired by the theoretical analysis of domain adaptation (Ben-David et al., 2010), a major focus has been reducing the discrepancy of representations between domains, so that a classifier that is learned on the source features works well on the target features. UDA methods can be categorized by the technique they use to measure this discrepancy. For example, (Long et al., 2013; Tzeng et al., 2014; 2015) use the maximum mean discrepancy (Gretton et al., 2012), (Sun & Saenko, 2016) use correlation alignment across domains, and domain adversarial neural networks (Ajakan et al., 2014; Ganin et al., 2016; Bousmalis et al., 2017; Saito et al., 2018) measure the domain discrepancy using a discriminator network. Maximum classifier discrepancy (Saito et al., 2018) (MCD) measures the domain discrepancy via multiple task classifiers, achieving SOTA performance.

**Semi-Supervised Learning (SSL).** In SSL, a portion of the training dataset is labeled and the remaining portion is unlabeled. SSL has seen great progress in recent years, including temporal ensemble (Laine & Aila, 2017), mean teacher (Tarvainen & Valpola, 2017), MixMatch (Berthelot et al., 2019), ReMixMatch (Berthelot et al., 2020), FixMatch (Sohn et al., 2020), and unsupervised data augmentation (Xie et al., 2019). While there is no technical barrier to applying SSL to UDA, only a few SSL methods have been applied to solve UDA problems; for example, on top of the discrepancy reduction techniques of UDA, several works (French et al., 2018; Long et al., 2018; Saito et al., 2019; Tran et al., 2019) propose to combine SSL techniques such as entropy minimization (Grandvalet & Bengio, 2005) or pseudo-labeling (Lee, 2013). While NoisyStudent (Xie et al., 2020) uses labeled data from ImageNet and unlabeled data from JFT in training, they do not leverage this distribution shift during training, and they only evaluate on the source domain (i.e., ImageNet).

**Semi-Supervised Domain Adaptation (SSDA)** has been studied in several settings, including vision (Donahue et al., 2013; Yao et al., 2015; Ao et al., 2017; Saito et al., 2019) and natural language processing (Jiang & Zhai, 2007; Daumé III et al., 2010; Guo & Xiao, 2012). Since SSDA assumes access to labeled data from multiple domains, early works have used separate models for each domain and regularized them with constraints (Donahue et al., 2013; Yao et al., 2015; Ao et al., 2017; Daumé III et al., 2010; Guo & Xiao, 2012). However, such methods are difficult to adapt to the UDA setting where labeled data is only available in a single domain. One recent exception is minimax entropy (MME) regularization (Saito et al., 2019), which can work in both the UDA and SSDA setting. However, unlike AdaMatch, MME requires a pre-trained network to work well.

**Transfer learning** is used to boost accuracy on small datasets by initializing model parameters with pre-trained weights first learned on a separate, larger dataset, which can compensate for the limited amount of labeled source data and boost the overall performance (Recht et al., 2019b; Kolesnikov et al., 2019). For example, standard experimental protocols on several UDA benchmarks, including Office-31 (Saenko et al., 2010), PACS (Li et al., 2017), and DomainNet (Peng et al., 2019), use ImageNet-pretrained models to initialize model parameters. Though useful for some cases in practice, it may not be the most general protocol to evaluate the advancement of UDA algorithms, especially in situations where no datasets exist for pre-training (for example, images with arbitrary number of

Figure 1: AdaMatch diagram illustrating the loss computations.

channels), or for domains other than vision, where no pre-training datasets exist. Thus, in this work, we mainly focus on evaluating methods under a non-transfer learning setting, which we consider to be more general. Although we also achieve state of the art results with transfer learning, we only present them for historical reasons and to illustrate that our method can use that setting too.

## 3 ADAMATCH

We now introduce AdaMatch, a new algorithm inspired by modern semi-supervised learning techniques, aimed at solving UDA, SSL, and SSDA. As typical in SSL, AdaMatch takes both an unlabeled and labeled dataset as input. We assume that the labeled data is drawn from a source domain while the unlabeled data is drawn from a target domain (for the SSL task, these domains are the same).

**Notation.** We use capital letters $X$, $Y$, $Z$ to denote minibatches of examples, labels and logits. Specifically, $X_{SL} \subset \mathbb{R}^{n_{SL} \times d}$ and $Y_{SL} \subset \{0,1\}^{n_{SL} \times k}$ denote the minibatch of source images and labels, respectively. Similarly, the minibatch of unlabeled target images is $X_{TU} \subset \mathbb{R}^{n_{TU} \times d}$. Here, $k$ is the number of classes and $d$ is the input dimension (for images $d = h \cdot w \cdot c$, where $h$ is height, $w$ is width, and $c$ is the number of channels). The minibatch size for the labeled data is $n_{SL}$ and the minibatch size of the unlabeled images is $n_{TU}$. Additionally, we use $Y^{(i)}$ to refer to its $i$-th row, and $Y^{(i,j)}$ to refer to the $i, j-$th element of $Y$. The model $f \colon \mathbb{R}^d \to \mathbb{R}^k$ takes images as input and outputs logits for each of $k$ classes. Importantly, the source and target domain are the same classification task, so the number of classes $k$ and the image dimension $d$ is the same for both domains.

### 3.1 METHOD DESCRIPTION

AdaMatch introduces three new techniques to account for differences between the source and target distributions – random logit interpolation, a relative confidence threshold, and a modified distribution alignment from ReMixMatch (Berthelot et al., 2020) – but builds upon the algorithmic backbone from FixMatch (Sohn et al., 2020). We first provide a high-level overview of the algorithm and then discuss the implementation details of the various components. For a brief summary of how each component helps with distribution shift and accompanying motivational examples, see Appendix A.

**Overview.** A high-level depiction of AdaMatch is in Figure 1. Two **augmentations** are made for each image: a weak and a strong one with the intent to make the class prediction harder on the strongly augmented image[2]. Next, we obtain logits by running two batches through the model: a batch of the source images and batch composed of both the source and target images. Each of the resulting batches of logits are influenced by their respective batch norm statistics, i.e. the source batch is only influenced by the source data batch norm statistics while the batch that combines source and target is influenced by both domains batch norm statistics. Two loss terms are then computed:

- The source loss term is responsible for predicting correct source labels and for aligning source and target logit domains. We first combine logits for the source images using **random logit interpolation**, which encourages the model to produce the same label for the hyperspace connecting source logits obtained from 1) only source examples and 2) a combination of source and target examples. In practice, this creates an implicit constraint to align the source and target domains in logit space. The newly obtained source logits are then used to compute the cross-entropy loss for the source data.

---

[2]We use the general "weak" and "strong" terms because they can be dependent on the task. For all of our results: Weak is shift and mirror about the $x$ axis. Strong is weak augmentation plus the addition of cutout.

- The target loss term is responsible for predicting the correct target labels and for aligning the target predictions to a desired class distribution. Since we don't assume access to labels for the target images, we create a pseudo-label for these images as follows. First, we rectify the class distribution obtained from weakly augmented target images to a desired class distribution using **distribution alignment**. If the target class distribution is known, it can be used directly. In the general case where it is not known, we use the source class distribution instead. We then select entries of the batch for which the rectified probabilities of the weakly augmented target image predictions are above a user-defined **confidence threshold**. A pseudo label is then made for these outputs by selecting the most confident class, and these pseudo-labels are used with a standard cross-entropy loss applied to the logits of the strongly augmented images.

**Augmentation.** For a dataset $D \in \{SL, TU\}$, we augment each image batch $X_D$ fed into AdaMatch twice, once with a *weak* augmentation and once with a *strong* augmentation, using the same types of weak and strong augmentations as (Berthelot et al., 2020). This forms a pair of batches $X_{D,w}$ and $X_{D,s}$ respectively, which we denote together as $X_D^{aug} = \{X_{D,w}, X_{D,s}\}$. From these pairs of batches, we then compute logits $Z'_{SL}, Z''_{SL}$ and $Z_{TU}$ as follows:

$$\{Z'_{SL}, Z_{TU}\} = f(\{X_{SL}^{aug}, X_{TU}^{aug}\}; \theta) \tag{1}$$

$$Z''_{SL} = f(X_{SL}^{aug}; \theta) \tag{2}$$

That is, we compute logits by calling the model twice for both the strong and weakly augmented images. The first time we pass both source and target inputs together in the same batch so the batch normalization statistics are shared, and the second time we only pass the source label data. We only update batch normalization statistics when computing $\{Z'_{SL}, Z_{TU}\} = f(\{X_{SL}^{aug}, X_{TU}^{aug}\}; \theta)$ to avoid double counting source labeled data. Note that without batch normalization, we would have $Z'_{SL} \equiv Z''_{SL}$. But batch normalization may make them slightly different.

**Random logit interpolation.** The role of random logit interpolation is to randomly combine the joint batch statistics from the source and target domains with the batch statistics from the source domain, which has the effect of producing batch statistics that are more representative of both domains. More precisely, during training, we obtain logits $Z_{SL}$ by randomly interpolating the logits $Z'_{SL}$ and $Z''_{SL}$ computed with different batch statistics:

$$Z_{SL} = \lambda \cdot Z'_{SL} + (1 - \lambda) \cdot Z''_{SL} \tag{3}$$

where we sample $\lambda \sim \mathcal{U}^{n_{SL} \cdot k}(0, 1)$. Note that each individual logit gets its own random factor.

Our underlying goal here is to minimize the loss for every point between $Z'_{SL}$ and $Z''_{SL}$, which can be accomplished by either 1) having $Z'_{SL}$ and $Z''_{SL}$ be equal to each other, or 2) having the whole line between $Z'_{SL}$ and $Z''_{SL}$ be a minima. Rather than picking one of the two ways, we formulate the problem as minimizing the loss for all connecting points, which gives the model the freedom to find the best possible solution. The key point is that we randomly choose the interpolation value , and then minimize the loss on this randomly interpolated value. Because we randomly choose the point on the line each time, over the course of training we can ensure that the entire line segment between $Z'_{SL}$ and $Z''_{SL}$ reaches low loss. Another way to achieve the same result would be to divide the interval into $N$ (a large number) of different segments, and then minimize the loss on all $N$ points. However this would be computationally expensive and so by randomly choosing a new each time we achieve the same result without increasing the computation cost by a factor of $N$.

**Distribution alignment.** Distribution alignment (Berthelot et al., 2020) can be seen as an additional form of model regularization that helps constrain the distribution of the class predictions to be more aligned with the true distribution. Without it, the classifier could just predict the most prevalent class or exhibit other failure modes. Ideally, if the target label distribution is known, we would use it directly. However, when the target label distribution is unknown, we approximate it using the only available distribution – the source label distribution. A limitation of this approach is that the more the source label distribution differs from the target distribution, the more incorrect the approximation will be, which may cause the model performance to degrade. However, in practice, we find that aligning the target pseudo-labels to match the source label distribution helps significantly.

Unlike ReMixMatch (Berthelot et al., 2020), we estimate the source label distribution from the *output* of the model rather than using the true labels. We make this change since the model may not be capable of matching the ground truth source label distribution (particularly when source accuracy is

low) but matching the source output distribution is a more attainable goal. To implement this, we first extract the logits for weakly and strongly augmented samples from the batch (by indexing):

$$Z_{SL} = \{Z_{SL,w}, Z_{SL,s}\} \qquad Z_{TU} = \{Z_{TU,w}, Z_{TU,s}\} \tag{4}$$

Then, we compute pseudo-labels for labeled sources and unlabeled targets:

$$\hat{Y}_{SL,w} = \texttt{softmax}(Z_{SL,w}) \in \mathbb{R}^{n_{SL} \cdot k} \qquad \hat{Y}_{TU,w} = \texttt{softmax}(Z_{TU,w}) \in \mathbb{R}^{n_{TU} \cdot k} \tag{5}$$

Using distribution alignment, we rectify the target unlabeled pseudo-labels by multiplying them by the ratio of the expected value of the weakly augmented source labels $\mathbb{E}[\hat{Y}_{SL,w}] \in \mathbb{R}^k$ to the expected value of the target labels $\mathbb{E}[\hat{Y}_{TU,w}] \in \mathbb{R}^k$, obtaining the final pseudo-labels $\tilde{Y}_{TU,w} \in \mathbb{R}^{n_{TU} \cdot k}$:

$$\tilde{Y}_{TU,w} = \texttt{normalize}\left(\hat{Y}_{TU,w} \frac{\mathbb{E}[\hat{Y}_{SL,w}]}{\mathbb{E}[\hat{Y}_{TU,w}]}\right) \tag{6}$$

$\texttt{normalize}$ ensures that the distribution still sums to 1. As could be seen $\mathbb{E}[\tilde{Y}_{TU,w}] = \mathbb{E}[\hat{Y}_{SL,w}]$, which confirms that distribution alignment makes the target pseudo-labels follow the source label distribution. If the target label distribution is known, one can simply replace the term $\mathbb{E}[\hat{Y}_{SL,w}]$ with it in the formula above.

**Relative confidence threshold.** A confidence threshold is typically used to select which predicted labels are confident enough to be used as pseudo-labels (Lee, 2013). However, since machine learning models are poorly calibrated (Guo et al., 2017), especially on out-of-distribution data Ovadia et al. (2019), the confidence varies from dataset to dataset depending on the ability of the model to learn its task. To address this issue, we introduce a relative confidence threshold which adjusts a user-provided confidence threshold relative to the confidence level of the classifier on the weakly augmented source data.

Specifically, we define the relative confidence threshold $c_\tau$ as the mean confidence of the top-1 prediction on the weakly augmented source data multiplied by a user provided threshold $\tau$:

$$c_\tau = \frac{\tau}{n_{SL}} \sum_{i=1}^{n_{SL}} \max_{j \in [1..k]} (\hat{Y}_{SL,w}^{(i,j)}) \tag{7}$$

We then compute a binary $mask \in \{0,1\}^{n_{TU}}$ by thresholding the weakly augmented target images with the relative confidence threshold $c_\tau$:

$$mask^{(i)} = \max_{j \in [1..k]} (\tilde{Y}_{TU,w}^{(i,j)}) \geq c_\tau \tag{8}$$

**Loss function**. The loss $\mathcal{L}(\theta)$ sums $\mathcal{L}_{\text{source}}(\theta)$ for the source and $\mathcal{L}_{\text{target}}(\theta)$ for the target.

$$\mathcal{L}_{\text{source}}(\theta) = \frac{1}{n_{SL}} \sum_{i=1}^{n_{SL}} H(Y_{SL}^{(i)}, Z_{SL,w}^{(i)}) + \frac{1}{n_{SL}} \sum_{i=1}^{n_{SL}} H(Y_{SL}^{(i)}, Z_{SL,s}^{(i)}) \tag{9}$$

$$\mathcal{L}_{\text{target}}(\theta) = \frac{1}{n_{TU}} \sum_{i=1}^{n_{TU}} H\left(\text{stop\_gradient}(\tilde{Y}_{TU,w}^{(i)}), Z_{TU,s}^{(i)}\right) \cdot mask^{(i)} \tag{10}$$

$$\mathcal{L}(\theta) = \mathcal{L}_{\text{source}}(\theta) + \mu(t)\mathcal{L}_{\text{target}}(\theta) \tag{11}$$

where $H(p,q) = -\sum p(x) \log q(x)$ is the cross-entropy loss and $\texttt{stop\_gradient}$ is a function that prevents gradient from back-propagating on its argument. Prevention of gradient back-propagation on guessed labels is a standard practice in SSL works that favors convergence. $\mu(t)$ is a warmup function that controls the unlabeled loss weight at every step of the training. The purpose is to shorten the convergence time for the model, and it should not significantly effect the model's final accuracy. In practice we use $\mu(t) = 1/2 - \cos(\min(\pi, 2\pi t/T))/2$ where $T$ is the total training steps. This particular function smoothly raises from 0 to 1 for the first half of the training and remains at 1 for the second half.

$\mathcal{L}_{\text{source}}(\theta)$ is the typical cross-entropy loss with the nuance that we call it twice: once on weakly augmented samples and once on strongly augmented ones. Similarly, $\mathcal{L}_{\text{target}}(\theta)$ is the masked cross-entropy loss, where entries for which the confidence is less than $c_\tau$ are zeroed. This loss term is exactly the same as in FixMatch (Sohn et al., 2020).

## 3.2 HYPER-PARAMETERS

AdaMatch only requires the following two hyper-parameters: (1) Confidence threshold $\tau$ (set to $0.9$ for all experiments). (2) Unlabeled target batch size ratio $uratio$ (set to 3 for all experiments) which defines how much larger is the unlabeled batch, e.g. $n_{tu} = n_{sl} \cdot uratio$.

## 3.3 EXTENSION TO SSL AND SSDA

SSL takes as input two types of data: labeled samples and their labels $X_L$ and unlabeled samples $X_U$. AdaMatch can be used without change for the SSL task by feeding $X_L$ in place of $X_{SL}$ and $X_U$ in place of $X_{TU}$. The same trick is provided for other methods in the experimental study.

SSDA differs from UDA by the presence of targeted labeled data $X_{TL}$. AdaMatch can be used without change by feeding the concatenated batch $X_{STL} = \{X_{SL}, X_{TL}\}$ in place of $X_{SL}$. As in SSL, the same trick works for other methods. Note there's a subtle effect on the $uratio$ which ends up being implicitly halved since the labeled batch is twice bigger in SSDA than in standard UDA.

## 4 EXPERIMENTAL SETUP

We evaluate AdaMatch on the SSL, UDA, and SSDA tasks using the standard DigitFive (Ganin et al., 2016) and DomainNet (Peng et al., 2019) visual domain adaptation benchmarks and we compare to various existing methods. DigitFive experiments and ablation studies were run on a single V100 GPU, other experiments were run on a single TPU.

**Datasets.** *Digit-Five* is composed of 5 domains, USPS (Hull, 1994), MNIST (LeCun et al., 1998), MNIST-M (Ganin et al., 2016), SVHN (Netzer et al., 2011), and synthetic numbers (Ganin et al., 2016). We resize all images into $32 \times 32^3$ and convert them to RGB. *DomainNet* (Peng et al., 2019) is composed of six domains from 345 object categories. Unlike prior work (Ganin et al., 2016; Saito et al., 2018; Peng et al., 2019), unless otherwise stated, we train models from scratch to focus on evaluating the efficacy of the algorithms themselves.

**Democratic research.** For DomainNet, we experiment with two image resolutions, $64 \times 64$ and $224 \times 224$. Though Peng et al. (2019) only evaluate with resolution $224 \times 224$, we include $64 \times 64$ to make future experiments using DomainNet more accessible to researchers with limited compute budget. Moreover, we find that AdaMatch can outperform the previous SOTA for DomainNet at $224 \times 224$ resolution even when trained with $64 \times 64$ resolution images.

**Network and training hyperparameters.** We use ResNetV2-101 (He et al., 2016) for resolution $224 \times 224$, WRN-34-2 (Zagoruyko & Komodakis, 2016) for $64 \times 64$, and WRN-28-2 for $32 \times 32$. We tried three confidence thresholds $(0.8, 0.9, 0.95)$ on $64 \times 64$ UDA clipart→infograph and picked the best. Additionally, we picked the largest $uratio$ allowed on 8 GPUs for a batch size of 64 using $224 \times 224$ images. We coarsely tuned (using 3 candidate values) the learning rate, learning rate decay, and weight decay using only the $64 \times 64$ supervised source data. For weight decay, we used a empirical rule: halving it for $32 \times 32$ neural networks and doubling it for the $224 \times 224$ to coarsely adapt the regularization to the capacity of the network (we used a weight decay of $0.0005$ for WRN-28-2, $0.001$ for WRN-34-2 and $0.002$ for ResNetV2-101). We set learning rate to $0.03$ and learning rate cosine decay to $0.25$. We trained DigitFive for 32M images, and DomainNet for 8M.

**Baselines.** We compare against popular SOTA methods from UDA and SSL, including maximum classifier discrepancy (MCD) (Saito et al., 2018), FixMatch with distribution alignment (FixMatch+), (Sohn et al., 2020), and NoisyStudent (Xie et al., 2019). For UDA, we include MCD since it is the current single-source SOTA on DomainNet (Peng et al., 2019). For SSL, we include FixMatch+ since it achieves SOTA performance on standard SSL benchmarks and our algorithm borrows many components from it. We also include NoisyStudent since it is a popular approach to SSL that achieves competitive results on large-scale datasets like ImageNet (Deng et al., 2009). Finally, we include our own baseline, BaselineBN, which uses fully supervised learning on the source domain but additionally feeds unlabeled target data through the network to update batch norm statistics (see Appendix B).

**Evaluation.** For each source and target dataset pair, we calculate the final accuracy as the median accuracy reported over the last ten checkpoints (we save a checkpoint every $2^{16}$ samples). In summary tables, we report the average final accuracy over all *target* datasets. Unlike SSL, most prior work

---

[3]Images of USPS are resized from $16 \times 16$ with bi-cubic interpolation. For MNIST, we pad an image with two zero-valued pixels on all sides.

in UDA evaluates using ImageNet pre-training. As explained in Section 2, we choose to report the majority of our results without pre-training. However, for easier comparison with prior work, we also evaluate ImageNet pre-trained AdaMatch, FixMatch+, and MCD for UDA in Section 5.1.1.

## 5  RESULTS

We now summarize our results for the UDA, SSL, and SSDA tasks. We ran *all five algorithms on all source-target pairs for every dataset* (20 for DigitFive and 30 for DomainNet64/224), varying the number for labels in SSL/SSDA tasks, for a comprehensive total of 2,420 experiments. This represents a major experimental undertaking and we hope that the results (especially the finer grained results on individual dataset pairs in Appendix E and the democratized setting) will serve future work.

### 5.1  UNSUPERVISED DOMAIN ADAPTATION (UDA)

| | DigitFive | DomainNet64 | DomainNet224 |
|---|---|---|---|
| BaselineBN | 62.5 | 15.8 | 18.9 |
| MCD | 52.1 | 14.9 | 14.9 |
| NoisyStudent | 73.3 | 21.4 | 23.9 |
| FixMatch+ | 95.6 | 20.1 | 20.8 |
| AdaMatch | **97.8** | **26.1** | **28.7** |
| Oracle | 98.8 | 60.5 | 65.9 |

Table 2: **For the UDA task, AdaMatch outperforms all other baseline algorithms on the DigitFive, DomainNet64, and DomainNet224 benchmarks**. For each algorithm and benchmark, we report the average *target* accuracy across all source→target pairs.

Table 2 shows the average target accuracy achieved by each algorithm for the UDA task on the DigitFive, DomainNet64, and DomainNet224 benchmarks. For the UDA task, we additionally compare to a fully-supervised learning setup that uses an oracle model that has access to all labels from both the source and target datasets (Oracle).

We find that AdaMatch outperforms all other algorithms we compare against, and the improvement is highest for the larger dataset size, DomainNet224, where AdaMatch achieves an average target accuracy of 28.7% compared to 23.9% for NoisyStudent. Additionally, on all three benchmark datasets, AdaMatch, FixMatch+, and NoisyStudent all significantly outperform both MCD and BaselineBN. This success illustrates that SSL methods can be applied out-of-the-box on UDA problems, but we can also significantly improve upon them if, as in AdaMatch, the distributional differences between the source and target data are accounted for.

For DomainNet64, we also observe that both AdaMatch and FixMatch+ *without pre-training on DomainNet64* are able to out-perform the previous state-of-the-art accuracy of 21.9% set by MCD *with pre-training on DomainNet224* (Peng et al., 2019). Overall, the accuracy gap between AdaMatch on DomainNet64 and DomainNet224 is relatively low – only 2% – and, for the purposes of democratizing research, we encourage future UDA research to evaluate on DomainNet64 since it is significantly less computationally intensive to run experiments on the smaller image size.

#### 5.1.1  UDA WITH PRE-TRAINING

| | random init (8M) | pre-train (8M) | pre-train (2M) | pre-train (max) |
|---|---|---|---|---|
| MCD | 14.9 | 22.3 | 21.9 | 22.7 |
| AdaMatch | **28.7** | **28.2** | **33.4** | **35.6** |

Table 3: **DomainNet224: In the pre-training UDA setting, AdaMatch also outperforms prior work.**. We report the target accuracy achieved after training for 8 million images (8M) (our standard training protocol), after early stopping at 2 million images (2M), and the maximum medium target accuracy (over a window of 10 checkpoints) achieved across the entire run (MAX).

In order to compare with prior UDA work which uses ImageNet pre-training, in Table 3 we evaluate AdaMatch versus MCD on the DomainNet224 benchmark when initializing from pre-trained ImageNet ResNet101 weights. As is typical with pre-training, we also found in our experiments that early stopping is necessary to get the best results for AdaMatch. Thus, for the pre-training results in Table 3, we report the final target accuracy achieved after training for 8 million images (our standard training protocol), the final accuracy achieved after early stopping at 2 million images, and the maximum median accuracy (over a window of 10 checkpoints) achieved across the entire run.

Using the standard training protocol of 8 million images, AdaMatch with pre-training is able to outperform MCD with pre-training, a difference of 28.2% versus 22.3%, respectively (we also confirm that the accuracy we are able to achieve on MCD closely matches the reported accuracy of 21.9% from (Peng et al., 2019)). However, we also observe that pre-trained AdaMatch peaks early on in training, and a significantly higher accuracy of 35.6% is achievable if an Oracle were to tell when to stop training. To account for this, we recommend early-stopping AdaMatch in the pre-trained setting.

## 5.2 Semi-Supervised Learning (SSL)

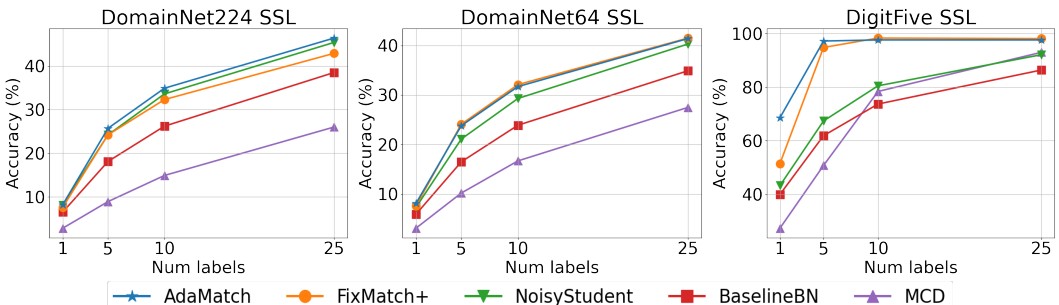

Figure 2: **Adamatch achieves state-of-the-art or competitive accuracy for the SSL task.** We evaluate on the DomainNet224, DomainNet64 and DigitFive benchmarks and vary the number of target labels. We report the average *target* accuracy across all source→target pairs. Average accuracy on the target datasets generally increases as we increase the number of target labels.

In the SSL setting, we only train on a single dataset (since there is no notion of source nor target) which we randomly divide into two groups: labeled and unlabeled. Figure 2 shows the average target accuracy for each algorithm on the SSL task as we vary the number of target labels (we also include the corresponding numerical results in Appendix D). We find that AdaMatch achieves state-of-the-art performance for DomainNet224, and competitive performance to FixMatch+ on the DigitFive and DomainNet64 benchmarks. Additionally, we observe that increasing the number of target labels generally results in better accuracy for all methods, and on the DomainNet224 dataset in particular, the gap in performance between AdaMatch and FixMatch+ widens as we increase the number labels.

Overall, AdaMatch's improvement over FixMatch+ is smaller on the SSL task compared to the SSDA or DA tasks, which is expected since AdaMatch by design is an extension of FixMatch+ to handle distribution shifts. This also suggests that the additional components of AdaMatch, i.e. random logit interpolation and relative confidence thresholding, improve accuracy more in settings where the unlabeled and labeled data are not drawn from the same distribution.

## 5.3 Semi-Supervised Domain Adaptation (SSDA)

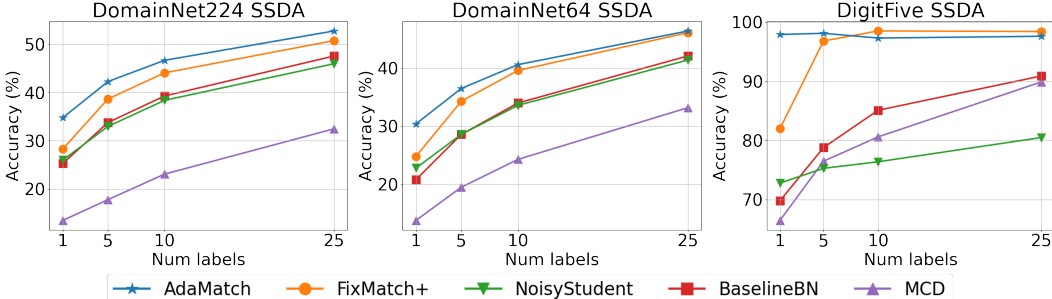

Figure 3: **Adamatch achieves state-of-the-art or competitive accuracy for the SSDA task.** We report the average *target* accuracy across all source→target pairs. Average accuracy on the target datasets generally increases as we increase the number of target labels.

In the SSDA setting, we randomly sample a subset of the target dataset and treat it as labeled. Figure 3 shows the performance of AdaMatch compared to all other algorithms on the SSDA task as we vary the number of target labels (corresponding numerical results are in Appendix D). AdaMatch

| Method | Accuracy |
|---|---|
| AdaMatch | 26.3 |
| w/o random logit interpolation | 25.2 |
| w/o distribution alignment | 17.7 |
| w/o relative confidence threshold | 23.2 |

Table 4: Ablation study on each component of AdaMatch on 6 UDA protocols of DomainNet. We evaluate algorithms by excluding the components one at a time.

outperforms all other algorithms on both DomainNet64 and DomainNet224. As expected, increasing the number of target labels improves the accuracy for all methods. In very low label regime, AdaMatch further improves its lead over other methods.

## 6 ABLATION STUDY

In this section, we perform an ablation analysis on each component of AdaMatch to better understand their importance and we provide a sensitivity analysis of hyperparameters, such as uratio, weight decay, confidence threshold, or augmentation strategies. We conduct our study on DomainNet using $64 \times 64$ as input and report the average accuracy across 6 domain adaptation protocols.[4]

**Exclude-One-Out Analysis.** AdaMatch improves upon FixMatch with several innovative techniques: random logit interpolation, adaptive confidence thresholding and distribution alignment (initially introduced by ReMixMatch and discussed in FixMatch as an extension but not per-se part of it). To better understand the role of each component, we conduct experiments by excluding one component at a time from AdaMatch. As we see in Table 4 all components contribute to the success of AdaMatch.

**Sensitivity Analysis.** AdaMatch only has two hyper-parameters to tune: uratio and confidence threshold. While it is not specific to AdaMatch, we also measure the sensitivity to weight decay as it was shown to be important to achieve a good performance in (Sohn et al., 2020). Results are in Figure 4. Similarly to the findings from FixMatch (Sohn et al., 2020), AdaMatch requires high confidence threshold. Also, we observe improved accuracy with higher $uratio$ at the expense of more compute. (As a reminder, $uratio$ defines the ratio of unlabeled to labeled data within a mini-batch, and not the ratio between the total number of unlabeled and labeled examples seen over the course of training.) For L2 weight decay, we find fairly reliable performance between $0.0001$ and $0.001$.

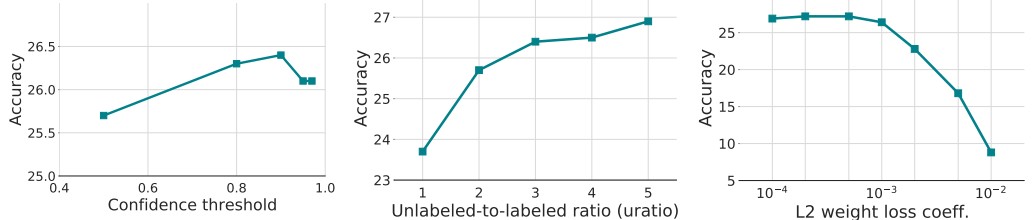

Figure 4: Ablation studies of AdaMatch on 6 DA protocols of DomainNet.

## 7 CONCLUSION

Machine learning models still suffer a drop in accuracy on out-of-distribution data. However, if we have access to unlabeled data from a shifted domain (the UDA setting) or even a small amount of labeled data from the shifted domain (the SSDA setting), we can greatly improve accuracy. In this work, we present AdaMatch, a general method designed to boost accuracy on domain shifts when given access to unlabeled data from the new domain. AdaMatch unifies the domains of UDA, SSL, and SSDA, demonstrating that one method can perform well at all three.

Overall, our work shows that it is possible to apply SSL algorithms out-of-the-box to the domain adaptation problem. By taking into account the distribution shift, AdaMatch can significantly improve upon SSL methods. This suggests that a promising direction for future work is to take new advances from SSL and translate or modify them for the domain adaptation setting. While AdaMatch outperforms prior work by a large margin, there is still an even larger margin left for improvement on out-of-distribution shifts. This leaves open an important question: which is more important for future progress, making use of unlabeled data more efficiently, or attempting to better model domain shifts?

---

[4]The 6 protocols include clipart→infograph, infograph→painting, painting→quickdraw, quickdraw→real, real→sketch, and sketch→clipart.

## ACKNOWLEDGEMENTS

We would like to thank Ekin Dogus Cubuk and Colin Raffel for helpful feedback on this work.

## ETHICS STATEMENT

AdaMatch is designed to make ML models robust to a domain shift between train and test time using a limited amount of labeled data and a large amount of unlabeled data. The development of data efficient ML algorithms are of utmost importance in democratization of ML methods. However, the confirmation bias of self-training (Arazo et al., 2019) could be a concern for building a fair ML model across major and minor classes. AdaMatch partially resolves the issue with the distribution alignment, but more in-depth investigation on the fairness and robustness of data-efficient ML algorithms should be done in the future.

## REPRODUCIBILITY

We wrote our experimental codes using the open-source Objax library (Objax Developers, 2020) and used the same hyperparameters for most of our experiments, which we specify in Section 4. Moreover, we released the open source code publicly at `https://github.com/google-research/adamatch`.

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

# A  MOTIVATION AND ILLUSTRATING EXAMPLES

We first briefly summarize the motivation for each of the components we introduce and explain how they help with distribution shift:

1. Random logit has the effect of producing batch statistics that are more representative of both domains (Section 3.1: Random Logit Interpolation) and creates an implicit constraint to align the source and target domains in logit space (Section 3.1: Overview).

2. Distribution alignment helps constrain the distribution of the class predictions to be more aligned with the true distribution (Section 3.1: Distribution Alignment). If we left it out, then the model would have no incentive to match the target distribution to the source distribution.

3. Relative confidence thresholding addresses the issue that models are poorly calibrated on out-of-distribution data. (Section 3.1: Relative Confidence Threshold) By including relative confidence thresholding, we can ensure that the target data is used as pseudo labels as often as it should be.

We next provide some illustrating examples to clarify the concepts of distribution alignment and relative confidence thresholding.

**Distribution alignment example**: Assume we have a source dataset that has two classes that should follow a frequency distribution of $\{0.4, 0.6\}$, e.g. 40% of samples are from class 1 and 60% from class 2. Let's say the model for the weakly augmented source data currently predicts an empirical frequency distribution of $\{0.3, 0.7\}$, e.g. 30% for class 1 and 70% for class 2. However, for the weakly augmented *target* data the model predicts on average $\{0.6, 0.4\}$.

We rectify a new prediction $P_{TU,w}$ by multiplying it elementwise by $\frac{\{0.3, 0.7\}}{\{0.6, 0.4\}}$, which has the effect of making the first class of $P_{TU,w}$ half as likely and the second class (roughly) twice as likely. In other words, it changes the class distribution on the weakly augmented unlabeled target domain from $\{0.6, 0.4\}$ to $\{0.3, 0.7\}$.

**Relative confidence thresholding example**: Confidence thresholding determines what is a confident pseudo-label for unlabeled data. However, since machine learning models are poorly calibrated (Guo et al., 2017), especially on out-of-distribution data, the confidence varies from dataset to dataset depending on the ability of the model to learn its task.

Suppose that we have a dataset X where a given model's average top-1 confidence is 0.7 on labeled data. Using a default confidence threshold of $\tau = 0.9$ for pseudo-labels will exclude almost all unlabeled data since they are unlikely to exceed the maximum labeled data confidence. In this scenario, relative confidence thresholding is particularly useful. By multiplying the default confidence threshold $\tau$ by the average top-1 labeled confidence, we can obtain a relative confidence ratio of $c_\tau = 0.9 \times 0.7 = 0.63$ which is more likely to capture a meaningful fraction of the unlabeled data.

In the case of CIFAR-10, typically the softmax of most if not all labeled training examples will reach 1.0. When the average top-1 confidence on the labeled data is 1.0, the relative confidence threshold $c_\tau$ and the default confidence threshold $\tau$ are the same. Therefore, one can see relative confidence thresholding as a generalization of the confidence thresholding concept.

# B  BASELINEBN

For BaselineBN, the same model and hyper-parameters are used, but the loss is computed differently. Both source and target are concatenated as a batch, and the loss is only computed on the (source) labeled logits for both weak and strong augmentations.

$$\{Z_{SL}, Z_{TU}\} = f(\{X_{SL}, X_{TU}\}; \theta)$$

$$\mathcal{L}_{\text{baseline}}(\theta) = \frac{1}{n_{SL}} \sum_{i=1}^{n_{SL}} H(Y_{SL}^{(i)}, Z_{SL,w}^{(i)}) + \frac{1}{n_{SL}} \sum_{i=1}^{n_{SL}} H(Y_{SL}^{(i)}, Z_{SL,s}^{(i)})$$

## C   UDA MODEL EVAL CURVES WITH AND WITHOUT PRE-TRAINED WEIGHTS

We randomly select a dataset pair from DomainNet224 (Sketch→Clipart) and plot the target accuracy versus the number of training images for both pre-trained and randomly initialized AdaMatch and MCD in Figure 5. For all pre-trained runs, we set weight decay to 0 and initialize model weights using an ImageNet pre-trained ResNet101 architecture.

Figure 5 shows that AdaMatch benefits significantly from early stopping. Although not plotted, we observe a similar pattern in other dataset pairs. Using a consistent early stopping rule of 2 million images, we found that, on average across all dataset pairs, AdaMatch with pre-training and early stopping outperforms randomly initialized AdaMatch by 4.7% (see Table 3).

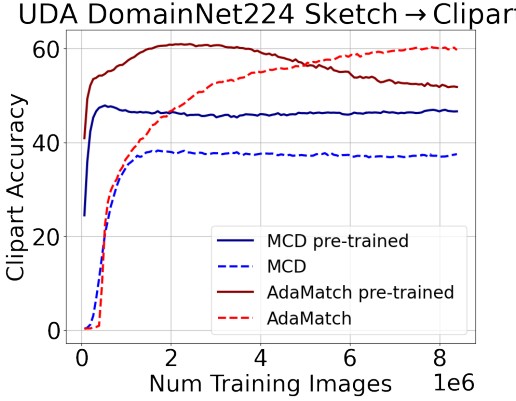

Figure 5: **When using pre-training, AdaMatch benefits significantly from early stopping**. We compare AdaMatch and MCD with and without pre-training in the UDA setting on a randomly selected dataset pair from DomainNet224, Sketch→Clipart.

# D SUMMARY RESULTS

In this section, we include the corresponding tables for the SSL and SSDA summary figures (Figure 2 and 3, respectively).

## D.1 SSL

| | DigitFive | | | | DomainNet64 | | | | DomainNet224 | | | | Avg |
|---|---|---|---|---|---|---|---|---|---|---|---|---|---|
| | 1 | 5 | 10 | 25 | 1 | 5 | 10 | 25 | 1 | 5 | 10 | 25 | |
| BaselineBN | 40.0 | 61.9 | 73.7 | 86.4 | 5.9 | 16.5 | 23.9 | 34.9 | 6.5 | 18.1 | 26.2 | 38.5 | 36.0 |
| MCD | 27.4 | 50.8 | 78.4 | 93.0 | 3.1 | 10.2 | 16.7 | 27.5 | 2.8 | 8.9 | 14.9 | 26.0 | 30.0 |
| NoisyStudent | 43.3 | 67.4 | 80.5 | 92.1 | 7.3 | 21.1 | 29.3 | 40.3 | 8.1 | 24.2 | 33.6 | 45.4 | 41.0 |
| FixMatch+ | 51.5 | 94.8 | 98.3 | 98.1 | 7.6 | 24.1 | 32.1 | 41.5 | 7.6 | 24.2 | 32.3 | 42.9 | 46.3 |
| AdaMatch | 68.6 | 97.2 | 97.6 | 97.6 | 8.1 | 23.8 | 31.7 | 41.4 | 8.2 | 25.7 | 34.9 | 46.4 | 48.4 |

Table 5: **SSL summary**. Adamatch achieves state-of-the-art or competitive accuracy for the SSL task. We evaluate on the DomainNet224, DomainNet64 and DigitFive benchmarks and vary the number of available labels. We report the average *target* accuracy across all source→target pairs. Average accuracy on the target datasets generally increases as we increase the number of target labels. See Figure 2 for the corresponding plot of accuracy versus number of target labels.

## D.2 SSDA

| | DigitFive | | | | DomainNet64 | | | | DomainNet224 | | | | Avg |
|---|---|---|---|---|---|---|---|---|---|---|---|---|---|
| | 1 | 5 | 10 | 25 | 1 | 5 | 10 | 25 | 1 | 5 | 10 | 25 | |
| BaselineBN | 69.8 | 78.8 | 85.1 | 90.9 | 20.8 | 28.6 | 34.0 | 42.1 | 25.3 | 33.8 | 39.3 | 47.6 | 49.7 |
| MCD | 66.5 | 76.5 | 80.6 | 89.9 | 13.8 | 19.5 | 24.3 | 33.2 | 13.5 | 17.8 | 23.1 | 32.5 | 40.9 |
| NoisyStudent | 72.8 | 75.3 | 76.4 | 80.5 | 22.8 | 28.6 | 33.6 | 41.4 | 26.0 | 33.0 | 38.4 | 46.0 | 47.9 |
| FixMatch+ | 82.0 | 96.8 | 98.5 | 98.4 | 24.8 | 34.2 | 39.6 | 46.1 | 28.3 | 38.7 | 44.1 | 50.8 | 56.9 |
| AdaMatch | 97.9 | 98.1 | 97.3 | 97.6 | 30.4 | 36.5 | 40.6 | 46.4 | 34.8 | 42.3 | 46.7 | 52.8 | 60.1 |

Table 6: **SSDA summary**. Adamatch achieves state-of-the-art or competitive accuracy for the SSDA task. We evaluate on the DomainNet224, DomainNet64 and DigitFive benchmarks and vary the number of target labels. We report the average *target* accuracy across all source→target pairs. Average accuracy on the target datasets generally increases as we increase the number of target labels. See Figure 3 for the corresponding plot of accuracy versus number of target labels.

# E INDIVIDUAL DATASET RESULTS

In this section, we evaluate all methods on the individual dataset pairs from the DigitFive, Domain-Net64, and DomainNet224 benchmarks for the UDA, SSL, and SSDA tasks.

| SSDA DigitFive AdaMatch (num_target_labels=1) | | | | | | |
|---|---|---|---|---|---|---|
| | mnist | mnistm | svhn | syndigit | usps | Avg |
| mnist | - | 99.1 | 96.7 | 99.4 | 97.8 | 98.3 |
| mnistm | 99.5 | - | 96.8 | 99.7 | 97.9 | 98.5 |
| svhn | 99.2 | 98.9 | - | 99.6 | 90.7 | 97.1 |
| syndigit | 99.3 | 98.9 | 97.0 | - | 97.1 | 98.1 |
| usps | 99.3 | 98.9 | 96.6 | 95.0 | - | 97.4 |
| Avg | 99.3 | 98.9 | 96.8 | 98.4 | 95.9 | 97.9 |

Table 7: Individual results for each dataset pair for DigitFive in the SSDA setting using AdaMatch.

| SSDA DigitFive AdaMatch (num_target_labels=5) | | | | | | |
|---|---|---|---|---|---|---|
| | mnist | mnistm | svhn | syndigit | usps | Avg |
| mnist | - | 99.2 | 96.7 | 99.5 | 97.8 | 98.3 |
| mnistm | 99.5 | - | 96.7 | 99.7 | 97.7 | 98.4 |
| svhn | 99.3 | 98.9 | - | 99.5 | 94.5 | 98.0 |
| syndigit | 99.3 | 98.9 | 96.8 | - | 97.0 | 98.0 |
| usps | 99.4 | 99.0 | 96.1 | 95.5 | - | 97.5 |
| Avg | 99.4 | 99.0 | 96.6 | 98.5 | 96.8 | 98.0 |

Table 8: Individual results for each dataset pair for DigitFive in the SSDA setting using AdaMatch.

| SSDA DigitFive AdaMatch (num_target_labels=10) | | | | | | |
|---|---|---|---|---|---|---|
| | mnist | mnistm | svhn | syndigit | usps | Avg |
| mnist | - | 99.2 | 96.7 | 99.7 | 97.7 | 98.3 |
| mnistm | 99.4 | - | 96.6 | 99.7 | 97.7 | 98.3 |
| svhn | 99.3 | 98.8 | - | 99.6 | 91.5 | 97.3 |
| syndigit | 99.3 | 99.0 | 96.9 | - | 97.4 | 98.2 |
| usps | 99.4 | 98.9 | 83.5 | 95.6 | - | 94.3 |
| Avg | 99.3 | 99.0 | 93.4 | 98.7 | 96.1 | 97.3 |

Table 9: Individual results for each dataset pair for DigitFive in the SSDA setting using AdaMatch.

| SSDA DigitFive AdaMatch (num_target_labels=25) | | | | | | |
|---|---|---|---|---|---|---|
| | mnist | mnistm | svhn | syndigit | usps | Avg |
| mnist | - | 99.2 | 96.1 | 99.4 | 97.9 | 98.2 |
| mnistm | 99.4 | - | 96.4 | 98.7 | 97.9 | 98.1 |
| svhn | 99.3 | 98.7 | - | 99.6 | 96.8 | 98.6 |
| syndigit | 99.3 | 98.8 | 96.4 | - | 97.6 | 98.0 |
| usps | 99.3 | 98.9 | 83.8 | 99.3 | - | 95.3 |
| Avg | 99.3 | 98.9 | 93.2 | 99.3 | 97.6 | 97.6 |

Table 10: Individual results for each dataset pair for DigitFive in the SSDA setting using AdaMatch.

**SSDA DigitFive BaselineBN (num_target_labels=1)**

|          | mnist | mnistm | svhn | syndigit | usps | Avg  |
| -------- | ----- | ------ | ---- | -------- | ---- | ---- |
| mnist    | -     | 72.8   | 42.1 | 78.3     | 95.8 | 72.2 |
| mnistm   | 97.1  | -      | 60.6 | 86.5     | 94.8 | 84.8 |
| svhn     | 76.5  | 37.0   | -    | 97.3     | 63.6 | 68.6 |
| syndigit | 85.7  | 31.1   | 77.0 | -        | 80.3 | 68.5 |
| usps     | 83.5  | 33.0   | 31.7 | 70.3     | -    | 54.6 |
| Avg      | 85.7  | 43.5   | 52.8 | 83.1     | 83.6 | 69.8 |

Table 11: Individual results for each dataset pair for DigitFive in the SSDA setting using BaselineBN.

**SSDA DigitFive BaselineBN (num_target_labels=5)**

|          | mnist | mnistm | svhn | syndigit | usps | Avg  |
| -------- | ----- | ------ | ---- | -------- | ---- | ---- |
| mnist    | -     | 79.7   | 59.4 | 90.0     | 96.4 | 81.4 |
| mnistm   | 96.5  | -      | 68.4 | 91.4     | 95.7 | 88.0 |
| svhn     | 90.6  | 49.1   | -    | 97.4     | 82.8 | 80.0 |
| syndigit | 91.5  | 50.8   | 80.0 | -        | 87.4 | 77.4 |
| usps     | 91.3  | 45.5   | 48.0 | 84.8     | -    | 67.4 |
| Avg      | 92.5  | 56.3   | 64.0 | 90.9     | 90.6 | 78.8 |

Table 12: Individual results for each dataset pair for DigitFive in the SSDA setting using BaselineBN.

**SSDA DigitFive BaselineBN (num_target_labels=10)**

|          | mnist | mnistm | svhn | syndigit | usps | Avg  |
| -------- | ----- | ------ | ---- | -------- | ---- | ---- |
| mnist    | -     | 85.6   | 68.3 | 92.2     | 96.9 | 85.8 |
| mnistm   | 98.2  | -      | 75.6 | 93.0     | 95.2 | 90.5 |
| svhn     | 94.8  | 65.3   | -    | 97.4     | 89.4 | 86.7 |
| syndigit | 95.9  | 68.1   | 83.5 | -        | 92.2 | 84.9 |
| usps     | 94.3  | 65.0   | 61.3 | 89.1     | -    | 77.4 |
| Avg      | 95.8  | 71.0   | 72.2 | 92.9     | 93.4 | 85.1 |

Table 13: Individual results for each dataset pair for DigitFive in the SSDA setting using BaselineBN.

**SSDA DigitFive BaselineBN (num_target_labels=25)**

|          | mnist | mnistm | svhn | syndigit | usps | Avg  |
| -------- | ----- | ------ | ---- | -------- | ---- | ---- |
| mnist    | -     | 93.2   | 76.8 | 95.8     | 97.1 | 90.7 |
| mnistm   | 98.8  | -      | 79.3 | 95.8     | 96.8 | 92.7 |
| svhn     | 96.8  | 83.2   | -    | 98.2     | 94.9 | 93.3 |
| syndigit | 97.0  | 84.7   | 86.4 | -        | 95.6 | 90.9 |
| usps     | 96.8  | 84.8   | 72.2 | 94.5     | -    | 87.1 |
| Avg      | 97.4  | 86.5   | 78.7 | 96.1     | 96.1 | 90.9 |

Table 14: Individual results for each dataset pair for DigitFive in the SSDA setting using BaselineBN.

| SSDA DigitFive FixMatch+ (num_target_labels=1) | | | | | | |
|---|---|---|---|---|---|---|
| | mnist | mnistm | svhn | syndigit | usps | Avg |
| mnist | - | 99.2 | 54.5 | 99.7 | 98.1 | 87.9 |
| mnistm | 99.4 | - | 54.7 | 99.7 | 97.8 | 87.9 |
| svhn | 97.9 | 10.9 | - | 99.6 | 97.3 | 76.4 |
| syndigit | 99.4 | 98.8 | 97.1 | - | 97.4 | 98.2 |
| usps | 99.3 | 9.8 | 28.8 | 99.7 | - | 59.4 |
| Avg | 99.0 | 54.7 | 58.8 | 99.7 | 97.7 | 82.0 |

Table 15: Individual results for each dataset pair for DigitFive in the SSDA setting using FixMatch+.

| SSDA DigitFive FixMatch+ (num_target_labels=5) | | | | | | |
|---|---|---|---|---|---|---|
| | mnist | mnistm | svhn | syndigit | usps | Avg |
| mnist | - | 99.2 | 80.8 | 99.7 | 98.0 | 94.4 |
| mnistm | 99.4 | - | 86.7 | 99.3 | 97.8 | 95.8 |
| svhn | 97.9 | 97.4 | - | 99.6 | 93.2 | 97.0 |
| syndigit | 99.3 | 98.9 | 96.9 | - | 97.7 | 98.2 |
| usps | 99.3 | 99.0 | 96.8 | 99.7 | - | 98.7 |
| Avg | 99.0 | 98.6 | 90.3 | 99.6 | 96.7 | 96.8 |

Table 16: Individual results for each dataset pair for DigitFive in the SSDA setting using FixMatch+.

| SSDA DigitFive FixMatch+ (num_target_labels=10) | | | | | | |
|---|---|---|---|---|---|---|
| | mnist | mnistm | svhn | syndigit | usps | Avg |
| mnist | - | 99.2 | 96.7 | 99.7 | 97.9 | 98.4 |
| mnistm | 99.5 | - | 97.0 | 99.8 | 97.7 | 98.5 |
| svhn | 99.3 | 98.8 | - | 99.6 | 97.3 | 98.8 |
| syndigit | 99.3 | 98.9 | 97.2 | - | 97.3 | 98.2 |
| usps | 99.4 | 98.9 | 96.8 | 99.7 | - | 98.7 |
| Avg | 99.4 | 98.9 | 96.9 | 99.7 | 97.6 | 98.5 |

Table 17: Individual results for each dataset pair for DigitFive in the SSDA setting using FixMatch+.

| SSDA DigitFive FixMatch+ (num_target_labels=25) | | | | | | |
|---|---|---|---|---|---|---|
| | mnist | mnistm | svhn | syndigit | usps | Avg |
| mnist | - | 99.2 | 96.5 | 99.8 | 97.8 | 98.3 |
| mnistm | 99.5 | - | 96.5 | 99.8 | 97.5 | 98.3 |
| svhn | 99.2 | 98.7 | - | 99.6 | 97.5 | 98.8 |
| syndigit | 99.4 | 98.8 | 96.6 | - | 97.5 | 98.1 |
| usps | 99.3 | 98.9 | 96.4 | 99.8 | - | 98.6 |
| Avg | 99.4 | 98.9 | 96.5 | 99.8 | 97.6 | 98.4 |

Table 18: Individual results for each dataset pair for DigitFive in the SSDA setting using FixMatch+.

**SSDA DigitFive MCD (num_target_labels=1)**

|          | mnist | mnistm | svhn | syndigit | usps | Avg  |
|----------|-------|--------|------|----------|------|------|
| mnist    | -     | 70.8   | 12.2 | 21.7     | 96.6 | 50.3 |
| mnistm   | 98.7  | -      | 7.7  | 87.9     | 94.3 | 72.2 |
| svhn     | 93.2  | 62.8   | -    | 98.0     | 92.4 | 86.6 |
| syndigit | 98.2  | 70.6   | 83.7 | -        | 94.9 | 86.8 |
| usps     | 98.3  | 10.0   | 10.8 | 27.9     | -    | 36.8 |
| Avg      | 97.1  | 53.5   | 28.6 | 58.9     | 94.5 | 66.5 |

Table 19: Individual results for each dataset pair for DigitFive in the SSDA setting using MCD.

**SSDA DigitFive MCD (num_target_labels=5)**

|          | mnist | mnistm | svhn | syndigit | usps | Avg  |
|----------|-------|--------|------|----------|------|------|
| mnist    | -     | 69.7   | 14.2 | 81.7     | 96.5 | 65.5 |
| mnistm   | 98.5  | -      | 7.6  | 88.3     | 95.1 | 72.4 |
| svhn     | 97.6  | 65.9   | -    | 98.1     | 91.5 | 88.3 |
| syndigit | 98.5  | 87.9   | 84.7 | -        | 94.9 | 91.5 |
| usps     | 97.9  | 61.0   | 9.8  | 91.1     | -    | 65.0 |
| Avg      | 98.1  | 71.1   | 29.1 | 89.8     | 94.5 | 76.5 |

Table 20: Individual results for each dataset pair for DigitFive in the SSDA setting using MCD.

**SSDA DigitFive MCD (num_target_labels=10)**

|          | mnist | mnistm | svhn | syndigit | usps | Avg  |
|----------|-------|--------|------|----------|------|------|
| mnist    | -     | 76.6   | 7.3  | 95.4     | 96.4 | 68.9 |
| mnistm   | 98.7  | -      | 13.5 | 95.9     | 95.5 | 75.9 |
| svhn     | 97.6  | 91.1   | -    | 98.3     | 94.2 | 95.3 |
| syndigit | 98.6  | 94.4   | 85.9 | -        | 95.3 | 93.5 |
| usps     | 98.3  | 73.2   | 10.7 | 95.3     | -    | 69.4 |
| Avg      | 98.3  | 83.8   | 29.4 | 96.2     | 95.4 | 80.6 |

Table 21: Individual results for each dataset pair for DigitFive in the SSDA setting using MCD.

**SSDA DigitFive MCD (num_target_labels=25)**

|          | mnist | mnistm | svhn | syndigit | usps | Avg  |
|----------|-------|--------|------|----------|------|------|
| mnist    | -     | 97.5   | 64.8 | 96.8     | 95.9 | 88.8 |
| mnistm   | 99.0  | -      | 23.7 | 97.2     | 95.6 | 78.9 |
| svhn     | 98.1  | 93.1   | -    | 98.3     | 95.0 | 96.1 |
| syndigit | 98.0  | 96.0   | 87.1 | -        | 95.3 | 94.1 |
| usps     | 98.5  | 93.5   | 77.0 | 97.0     | -    | 91.5 |
| Avg      | 98.4  | 95.0   | 63.1 | 97.3     | 95.5 | 89.9 |

Table 22: Individual results for each dataset pair for DigitFive in the SSDA setting using MCD.

| SSDA DigitFive NoisyStudent (num_target_labels=1) | | | | | | |
|---|---|---|---|---|---|---|
| | mnist | mnistm | svhn | syndigit | usps | Avg |
| mnist | - | 32.8 | 59.7 | 71.8 | 94.0 | 64.6 |
| mnistm | 99.2 | - | 58.6 | 88.0 | 95.1 | 85.2 |
| svhn | 80.2 | 44.2 | - | 98.8 | 85.9 | 77.3 |
| syndigit | 91.8 | 51.4 | 88.8 | - | 88.1 | 80.0 |
| usps | 95.5 | 25.3 | 33.4 | 73.3 | - | 56.9 |
| Avg | 91.7 | 38.4 | 60.1 | 83.0 | 90.8 | 72.8 |

Table 23: Individual results for each dataset pair for DigitFive in the SSDA setting using NoisyStudent.

| SSDA DigitFive NoisyStudent (num_target_labels=5) | | | | | | |
|---|---|---|---|---|---|---|
| | mnist | mnistm | svhn | syndigit | usps | Avg |
| mnist | - | 28.1 | 63.1 | 88.6 | 95.0 | 68.7 |
| mnistm | 99.2 | - | 64.9 | 90.1 | 93.1 | 86.8 |
| svhn | 80.7 | 49.6 | - | 98.9 | 85.3 | 78.6 |
| syndigit | 92.5 | 62.3 | 89.3 | - | 86.6 | 82.7 |
| usps | 94.2 | 30.4 | 38.2 | 76.7 | - | 59.9 |
| Avg | 91.6 | 42.6 | 63.9 | 88.6 | 90.0 | 75.3 |

Table 24: Individual results for each dataset pair for DigitFive in the SSDA setting using NoisyStudent.

| SSDA DigitFive NoisyStudent (num_target_labels=10) | | | | | | |
|---|---|---|---|---|---|---|
| | mnist | mnistm | svhn | syndigit | usps | Avg |
| mnist | - | 26.8 | 59.9 | 86.8 | 96.9 | 67.6 |
| mnistm | 99.3 | - | 62.8 | 88.5 | 95.7 | 86.6 |
| svhn | 78.5 | 58.0 | - | 98.9 | 88.8 | 81.0 |
| syndigit | 92.5 | 70.9 | 89.4 | - | 87.9 | 85.2 |
| usps | 94.2 | 31.8 | 41.4 | 78.6 | - | 61.5 |
| Avg | 91.1 | 46.9 | 63.4 | 88.2 | 92.3 | 76.4 |

Table 25: Individual results for each dataset pair for DigitFive in the SSDA setting using NoisyStudent.

| SSDA DigitFive NoisyStudent (num_target_labels=25) | | | | | | |
|---|---|---|---|---|---|---|
| | mnist | mnistm | svhn | syndigit | usps | Avg |
| mnist | - | 32.2 | 66.7 | 87.6 | 96.9 | 70.8 |
| mnistm | 99.2 | - | 79.3 | 91.7 | 95.6 | 91.4 |
| svhn | 86.6 | 68.9 | - | 98.8 | 89.2 | 85.9 |
| syndigit | 93.4 | 75.3 | 90.5 | - | 89.2 | 87.1 |
| usps | 94.9 | 62.7 | 34.0 | 78.0 | - | 67.4 |
| Avg | 93.5 | 59.8 | 67.6 | 89.0 | 92.7 | 80.5 |

Table 26: Individual results for each dataset pair for DigitFive in the SSDA setting using NoisyStudent.

| SSDA DomainNet224 AdaMatch (num_target_labels=1) | | | | | | | |
|---|---|---|---|---|---|---|---|
| | clipart | infograph | painting | quickdraw | real | sketch | Avg |
| clipart | - | 14.7 | 35.0 | 34.5 | 47.3 | 47.0 | 35.7 |
| infograph | 37.9 | - | 26.9 | 22.8 | 37.8 | 29.9 | 31.1 |
| painting | 48.9 | 15.5 | - | 28.5 | 51.0 | 43.3 | 37.4 |
| quickdraw | 37.0 | 2.9 | 13.8 | - | 27.8 | 24.1 | 21.1 |
| real | 59.6 | 17.9 | 47.9 | 36.5 | - | 45.3 | 41.4 |
| sketch | 61.5 | 16.4 | 42.2 | 39.0 | 50.4 | - | 41.9 |
| Avg | 49.0 | 13.5 | 33.2 | 32.3 | 42.9 | 37.9 | 34.8 |

Table 27: Individual results for each dataset pair for DomainNet224 in the SSDA setting using AdaMatch.

| SSDA DomainNet224 AdaMatch (num_target_labels=5) | | | | | | | |
|---|---|---|---|---|---|---|---|
| | clipart | infograph | painting | quickdraw | real | sketch | Avg |
| clipart | - | 15.6 | 38.6 | 50.9 | 53.0 | 50.1 | 41.6 |
| infograph | 48.1 | - | 32.9 | 43.1 | 46.2 | 39.3 | 41.9 |
| painting | 55.0 | 16.6 | - | 44.8 | 56.0 | 48.3 | 44.1 |
| quickdraw | 51.3 | 6.8 | 26.5 | - | 43.0 | 36.4 | 32.8 |
| real | 65.0 | 19.4 | 49.8 | 51.7 | - | 50.4 | 47.3 |
| sketch | 64.2 | 17.4 | 44.5 | 49.9 | 54.7 | - | 46.1 |
| Avg | 56.7 | 15.2 | 38.5 | 48.1 | 50.6 | 44.9 | 42.3 |

Table 28: Individual results for each dataset pair for DomainNet224 in the SSDA setting using AdaMatch.

| SSDA DomainNet224 AdaMatch (num_target_labels=10) | | | | | | | |
|---|---|---|---|---|---|---|---|
| | clipart | infograph | painting | quickdraw | real | sketch | Avg |
| clipart | - | 17.6 | 41.9 | 55.6 | 57.5 | 53.8 | 45.3 |
| infograph | 53.2 | - | 37.2 | 50.3 | 52.5 | 45.2 | 47.7 |
| painting | 59.4 | 17.8 | - | 52.6 | 59.9 | 51.2 | 48.2 |
| quickdraw | 56.7 | 10.0 | 34.6 | - | 51.0 | 44.5 | 39.4 |
| real | 67.6 | 20.9 | 51.8 | 56.9 | - | 54.3 | 50.3 |
| sketch | 66.4 | 19.4 | 47.2 | 55.8 | 58.6 | - | 49.5 |
| Avg | 60.7 | 17.1 | 42.5 | 54.2 | 55.9 | 49.8 | 46.7 |

Table 29: Individual results for each dataset pair for DomainNet224 in the SSDA setting using AdaMatch.

| SSDA DomainNet224 AdaMatch (num_target_labels=25) | | | | | | | |
|---|---|---|---|---|---|---|---|
| | clipart | infograph | painting | quickdraw | real | sketch | Avg |
| clipart | - | 20.8 | 49.2 | 62.0 | 64.2 | 58.1 | 50.9 |
| infograph | 61.5 | - | 44.9 | 58.4 | 60.9 | 51.5 | 55.4 |
| painting | 65.3 | 21.0 | - | 59.5 | 66.2 | 56.9 | 53.8 |
| quickdraw | 64.1 | 14.8 | 44.5 | - | 61.1 | 52.6 | 47.4 |
| real | 71.1 | 24.2 | 56.1 | 62.9 | - | 58.6 | 54.6 |
| sketch | 70.6 | 22.7 | 52.4 | 61.9 | 64.9 | - | 54.5 |
| Avg | 66.5 | 20.7 | 49.4 | 60.9 | 63.5 | 55.5 | 52.8 |

Table 30: Individual results for each dataset pair for DomainNet224 in the SSDA setting using AdaMatch.

| SSDA DomainNet224 BaselineBN (num_target_labels=1) | | | | | | | |
|---|---|---|---|---|---|---|---|
| | clipart | infograph | painting | quickdraw | real | sketch | Avg |
| clipart | - | 11.7 | 25.2 | 22.5 | 36.4 | 38.2 | 26.8 |
| infograph | 24.9 | - | 17.1 | 14.3 | 24.3 | 20.9 | 20.3 |
| painting | 39.7 | 12.2 | - | 17.7 | 44.2 | 34.1 | 29.6 |
| quickdraw | 24.7 | 2.4 | 7.4 | - | 15.2 | 15.8 | 13.1 |
| real | 47.9 | 14.7 | 40.4 | 20.2 | - | 35.1 | 31.7 |
| sketch | 51.4 | 12.1 | 31.4 | 20.5 | 37.4 | - | 30.6 |
| Avg | 37.7 | 10.6 | 24.3 | 19.0 | 31.5 | 28.8 | 25.3 |

Table 31: Individual results for each dataset pair for DomainNet224 in the SSDA setting using BaselineBN.

| SSDA DomainNet224 BaselineBN (num_target_labels=5) | | | | | | | |
|---|---|---|---|---|---|---|---|
| | clipart | infograph | painting | quickdraw | real | sketch | Avg |
| clipart | - | 12.8 | 29.8 | 39.8 | 42.9 | 43.2 | 33.7 |
| infograph | 38.9 | - | 22.2 | 33.7 | 34.5 | 29.9 | 31.8 |
| painting | 49.4 | 13.7 | - | 36.0 | 50.2 | 40.4 | 37.9 |
| quickdraw | 38.6 | 5.2 | 15.8 | - | 27.4 | 25.9 | 22.6 |
| real | 56.0 | 15.8 | 43.0 | 37.9 | - | 41.6 | 38.9 |
| sketch | 57.6 | 13.8 | 34.7 | 38.5 | 43.6 | - | 37.6 |
| Avg | 48.1 | 12.3 | 29.1 | 37.2 | 39.7 | 36.2 | 33.8 |

Table 32: Individual results for each dataset pair for DomainNet224 in the SSDA setting using BaselineBN.

| SSDA DomainNet224 BaselineBN (num_target_labels=10) | | | | | | | |
|---|---|---|---|---|---|---|---|
| | clipart | infograph | painting | quickdraw | real | sketch | Avg |
| clipart | - | 14.5 | 33.9 | 47.5 | 48.6 | 47.5 | 38.4 |
| infograph | 47.3 | - | 27.1 | 43.6 | 41.8 | 36.7 | 39.3 |
| painting | 55.0 | 15.5 | - | 45.1 | 54.2 | 45.2 | 43.0 |
| quickdraw | 46.1 | 7.8 | 21.9 | - | 37.1 | 33.9 | 29.4 |
| real | 59.9 | 18.0 | 45.7 | 46.3 | - | 46.9 | 43.4 |
| sketch | 61.0 | 15.7 | 38.3 | 47.3 | 49.8 | - | 42.4 |
| Avg | 53.9 | 14.3 | 33.4 | 46.0 | 46.3 | 42.0 | 39.3 |

Table 33: Individual results for each dataset pair for DomainNet224 in the SSDA setting using BaselineBN.

| SSDA DomainNet224 BaselineBN (num_target_labels=25) | | | | | | | |
|---|---|---|---|---|---|---|---|
| | clipart | infograph | painting | quickdraw | real | sketch | Avg |
| clipart | - | 18.1 | 43.0 | 56.5 | 57.0 | 53.8 | 45.7 |
| infograph | 59.5 | - | 38.3 | 54.0 | 53.8 | 47.7 | 50.7 |
| painting | 63.5 | 18.9 | - | 55.3 | 60.9 | 52.5 | 50.2 |
| quickdraw | 57.3 | 12.9 | 35.2 | - | 49.6 | 44.8 | 40.0 |
| real | 66.3 | 21.3 | 51.2 | 56.0 | - | 53.9 | 49.7 |
| sketch | 67.0 | 20.4 | 46.2 | 56.0 | 57.9 | - | 49.5 |
| Avg | 62.7 | 18.3 | 42.8 | 55.6 | 55.8 | 50.5 | 47.6 |

Table 34: Individual results for each dataset pair for DomainNet224 in the SSDA setting using BaselineBN.

| SSDA DomainNet224 FixMatch+ (num_target_labels=1) | | | | | | | |
| --- | --- | --- | --- | --- | --- | --- | --- |
| | clipart | infograph | painting | quickdraw | real | sketch | Avg |
| clipart | - | 12.5 | 28.6 | 30.2 | 41.3 | 41.6 | 30.8 |
| infograph | 27.8 | - | 17.4 | 17.8 | 25.9 | 22.0 | 22.2 |
| painting | 43.6 | 12.2 | - | 23.3 | 45.9 | 35.1 | 32.0 |
| quickdraw | 35.3 | 2.5 | 10.5 | - | 19.7 | 18.3 | 17.3 |
| real | 51.2 | 14.4 | 40.9 | 25.9 | - | 37.2 | 33.9 |
| sketch | 54.9 | 12.7 | 33.1 | 28.2 | 39.6 | - | 33.7 |
| Avg | 42.6 | 10.9 | 26.1 | 25.1 | 34.5 | 30.8 | 28.3 |

Table 35: Individual results for each dataset pair for DomainNet224 in the SSDA setting using FixMatch+.

| SSDA DomainNet224 FixMatch+ (num_target_labels=5) | | | | | | | |
| --- | --- | --- | --- | --- | --- | --- | --- |
| | clipart | infograph | painting | quickdraw | real | sketch | Avg |
| clipart | - | 13.9 | 35.2 | 48.8 | 49.1 | 48.1 | 39.0 |
| infograph | 46.5 | - | 24.5 | 41.7 | 39.0 | 34.3 | 37.2 |
| painting | 54.7 | 13.7 | - | 44.6 | 51.9 | 43.4 | 41.7 |
| quickdraw | 51.2 | 5.7 | 22.8 | - | 36.8 | 34.1 | 30.1 |
| real | 60.4 | 15.6 | 44.4 | 45.9 | - | 44.4 | 42.1 |
| sketch | 61.9 | 14.1 | 38.5 | 47.9 | 48.2 | - | 42.1 |
| Avg | 54.9 | 12.6 | 33.1 | 45.8 | 45.0 | 40.9 | 38.7 |

Table 36: Individual results for each dataset pair for DomainNet224 in the SSDA setting using FixMatch+.

| SSDA DomainNet224 FixMatch+ (num_target_labels=10) | | | | | | | |
| --- | --- | --- | --- | --- | --- | --- | --- |
| | clipart | infograph | painting | quickdraw | real | sketch | Avg |
| clipart | - | 16.0 | 40.3 | 55.3 | 54.6 | 51.8 | 43.6 |
| infograph | 55.1 | - | 30.7 | 50.5 | 46.6 | 41.5 | 44.9 |
| painting | 60.3 | 15.4 | - | 52.6 | 56.4 | 48.6 | 46.7 |
| quickdraw | 56.5 | 8.9 | 31.1 | - | 46.1 | 41.4 | 36.8 |
| real | 64.2 | 18.0 | 47.0 | 53.3 | - | 49.6 | 46.4 |
| sketch | 64.8 | 16.0 | 42.7 | 54.3 | 54.4 | - | 46.4 |
| Avg | 60.2 | 14.9 | 38.4 | 53.2 | 51.6 | 46.6 | 44.1 |

Table 37: Individual results for each dataset pair for DomainNet224 in the SSDA setting using FixMatch+.

| SSDA DomainNet224 FixMatch+ (num_target_labels=25) | | | | | | | |
| --- | --- | --- | --- | --- | --- | --- | --- |
| | clipart | infograph | painting | quickdraw | real | sketch | Avg |
| clipart | - | 19.2 | 47.9 | 60.8 | 61.6 | 56.7 | 49.2 |
| infograph | 64.3 | - | 41.7 | 58.4 | 57.0 | 51.3 | 54.5 |
| painting | 66.5 | 18.4 | - | 59.2 | 62.3 | 54.8 | 52.2 |
| quickdraw | 63.7 | 13.5 | 42.9 | - | 56.5 | 50.1 | 45.3 |
| real | 69.1 | 21.2 | 50.2 | 60.2 | - | 56.3 | 51.4 |
| sketch | 69.2 | 20.1 | 49.8 | 60.5 | 61.4 | - | 52.2 |
| Avg | 66.6 | 18.5 | 46.5 | 59.8 | 59.8 | 53.8 | 50.8 |

Table 38: Individual results for each dataset pair for DomainNet224 in the SSDA setting using FixMatch+.

| SSDA DomainNet224 MCD (num_target_labels=1) | | | | | | | |
|---|---|---|---|---|---|---|---|
| | clipart | infograph | painting | quickdraw | real | sketch | Avg |
| clipart | - | 5.5 | 11.2 | 16.0 | 22.0 | 18.8 | 14.7 |
| infograph | 9.1 | - | 6.1 | 7.2 | 9.7 | 4.6 | 7.3 |
| painting | 20.8 | 5.5 | - | 9.6 | 27.4 | 13.8 | 15.4 |
| quickdraw | 8.4 | 1.0 | 1.5 | - | 3.8 | 4.3 | 3.8 |
| real | 35.8 | 9.4 | 27.2 | 12.6 | - | 19.8 | 21.0 |
| sketch | 32.5 | 6.2 | 15.2 | 17.6 | 22.0 | - | 18.7 |
| Avg | 21.3 | 5.5 | 12.2 | 12.6 | 17.0 | 12.3 | 13.5 |

Table 39: Individual results for each dataset pair for DomainNet224 in the SSDA setting using MCD.

| SSDA DomainNet224 MCD (num_target_labels=5) | | | | | | | |
|---|---|---|---|---|---|---|---|
| | clipart | infograph | painting | quickdraw | real | sketch | Avg |
| clipart | - | 5.8 | 12.4 | 29.2 | 24.6 | 23.0 | 19.0 |
| infograph | 13.9 | - | 7.1 | 22.5 | 13.5 | 7.1 | 12.8 |
| painting | 23.7 | 5.8 | - | 25.7 | 29.0 | 15.3 | 19.9 |
| quickdraw | 16.9 | 1.9 | 3.0 | - | 9.9 | 9.6 | 8.3 |
| real | 38.6 | 9.7 | 26.5 | 26.4 | - | 22.3 | 24.7 |
| sketch | 36.3 | 6.1 | 15.6 | 29.0 | 25.0 | - | 22.4 |
| Avg | 25.9 | 5.9 | 12.9 | 26.6 | 20.4 | 15.5 | 17.8 |

Table 40: Individual results for each dataset pair for DomainNet224 in the SSDA setting using MCD.

| SSDA DomainNet224 MCD (num_target_labels=10) | | | | | | | |
|---|---|---|---|---|---|---|---|
| | clipart | infograph | painting | quickdraw | real | sketch | Avg |
| clipart | - | 7.0 | 14.3 | 37.8 | 30.5 | 28.2 | 23.6 |
| infograph | 20.8 | - | 9.2 | 32.0 | 19.2 | 15.5 | 19.3 |
| painting | 29.3 | 6.8 | - | 33.2 | 33.3 | 24.1 | 25.3 |
| quickdraw | 26.0 | 2.9 | 5.8 | - | 16.8 | 17.6 | 13.8 |
| real | 42.9 | 10.8 | 28.9 | 34.5 | - | 27.8 | 29.0 |
| sketch | 42.5 | 7.0 | 17.5 | 38.4 | 31.2 | - | 27.3 |
| Avg | 32.3 | 6.9 | 15.1 | 35.2 | 26.2 | 22.6 | 23.1 |

Table 41: Individual results for each dataset pair for DomainNet224 in the SSDA setting using MCD.

| SSDA DomainNet224 MCD (num_target_labels=25) | | | | | | | |
|---|---|---|---|---|---|---|---|
| | clipart | infograph | painting | quickdraw | real | sketch | Avg |
| clipart | - | 10.3 | 22.6 | 46.7 | 41.2 | 39.5 | 32.1 |
| infograph | 34.2 | - | 16.9 | 43.5 | 30.8 | 29.9 | 31.1 |
| painting | 42.6 | 8.9 | - | 45.7 | 42.7 | 37.9 | 35.6 |
| quickdraw | 40.4 | 5.3 | 15.4 | - | 30.5 | 30.6 | 24.4 |
| real | 50.1 | 12.9 | 34.3 | 46.2 | - | 38.4 | 36.4 |
| sketch | 50.6 | 11.2 | 28.0 | 47.0 | 41.5 | - | 35.7 |
| Avg | 43.6 | 9.7 | 23.4 | 45.8 | 37.3 | 35.3 | 32.5 |

Table 42: Individual results for each dataset pair for DomainNet224 in the SSDA setting using MCD.

| SSDA DomainNet224 NoisyStudent (num_target_labels=1) | | | | | | | |
|---|---|---|---|---|---|---|---|
| | clipart | infograph | painting | quickdraw | real | sketch | Avg |
| clipart | - | 12.1 | 28.8 | 24.9 | 39.8 | 42.6 | 29.6 |
| infograph | 27.5 | - | 19.9 | 8.1 | 25.5 | 22.1 | 20.6 |
| painting | 40.9 | 11.0 | - | 9.7 | 43.9 | 33.5 | 27.8 |
| quickdraw | 25.8 | 2.0 | 6.1 | - | 12.7 | 15.9 | 12.5 |
| real | 48.1 | 13.1 | 41.1 | 14.4 | - | 34.0 | 30.1 |
| sketch | 56.5 | 13.7 | 37.1 | 27.0 | 43.0 | - | 35.5 |
| Avg | 39.8 | 10.4 | 26.6 | 16.8 | 33.0 | 29.6 | 26.0 |

Table 43: Individual results for each dataset pair for DomainNet224 in the SSDA setting using NoisyStudent.

| SSDA DomainNet224 NoisyStudent (num_target_labels=5) | | | | | | | |
|---|---|---|---|---|---|---|---|
| | clipart | infograph | painting | quickdraw | real | sketch | Avg |
| clipart | - | 13.3 | 31.9 | 36.4 | 44.3 | 45.8 | 34.3 |
| infograph | 39.7 | - | 25.4 | 26.5 | 36.0 | 29.9 | 31.5 |
| painting | 48.7 | 12.9 | - | 24.5 | 49.9 | 38.0 | 34.8 |
| quickdraw | 35.1 | 4.3 | 15.0 | - | 25.8 | 24.2 | 20.9 |
| real | 54.8 | 14.4 | 43.5 | 32.7 | - | 40.6 | 37.2 |
| sketch | 59.2 | 14.4 | 39.4 | 37.5 | 46.9 | - | 39.5 |
| Avg | 47.5 | 11.9 | 31.0 | 31.5 | 40.6 | 35.7 | 33.0 |

Table 44: Individual results for each dataset pair for DomainNet224 in the SSDA setting using NoisyStudent.

| SSDA DomainNet224 NoisyStudent (num_target_labels=10) | | | | | | | |
|---|---|---|---|---|---|---|---|
| | clipart | infograph | painting | quickdraw | real | sketch | Avg |
| clipart | - | 15.3 | 36.5 | 44.4 | 48.5 | 49.2 | 38.8 |
| infograph | 47.9 | - | 29.8 | 38.3 | 41.0 | 37.0 | 38.8 |
| painting | 53.7 | 14.7 | - | 35.8 | 52.6 | 42.6 | 39.9 |
| quickdraw | 43.4 | 6.5 | 20.8 | - | 34.6 | 32.3 | 27.5 |
| real | 58.5 | 16.4 | 45.9 | 43.8 | - | 45.1 | 41.9 |
| sketch | 62.1 | 16.9 | 42.1 | 45.2 | 51.7 | - | 43.6 |
| Avg | 53.1 | 14.0 | 35.0 | 41.5 | 45.7 | 41.2 | 38.4 |

Table 45: Individual results for each dataset pair for DomainNet224 in the SSDA setting using NoisyStudent.

| SSDA DomainNet224 NoisyStudent (num_target_labels=25) | | | | | | | |
|---|---|---|---|---|---|---|---|
| | clipart | infograph | painting | quickdraw | real | sketch | Avg |
| clipart | - | 18.8 | 43.0 | 53.0 | 55.3 | 53.7 | 44.8 |
| infograph | 58.0 | - | 39.4 | 51.6 | 50.8 | 46.4 | 49.2 |
| painting | 61.5 | 17.9 | - | 49.9 | 57.3 | 49.6 | 47.2 |
| quickdraw | 54.5 | 11.8 | 33.7 | - | 47.2 | 42.6 | 38.0 |
| real | 64.0 | 20.0 | 49.6 | 54.0 | - | 51.6 | 47.8 |
| sketch | 65.9 | 20.3 | 48.4 | 53.8 | 55.7 | - | 48.8 |
| Avg | 60.8 | 17.8 | 42.8 | 52.5 | 53.3 | 48.8 | 46.0 |

Table 46: Individual results for each dataset pair for DomainNet224 in the SSDA setting using NoisyStudent.

| SSDA DomainNet64 AdaMatch (num_target_labels=1) | | | | | | | |
|---|---|---|---|---|---|---|---|
| | clipart | infograph | painting | quickdraw | real | sketch | Avg |
| clipart | - | 11.7 | 28.1 | 37.1 | 40.7 | 39.7 | 31.5 |
| infograph | 29.2 | - | 20.5 | 23.3 | 31.0 | 24.1 | 25.6 |
| painting | 42.8 | 12.6 | - | 26.5 | 44.7 | 36.4 | 32.6 |
| quickdraw | 35.7 | 2.6 | 13.2 | - | 25.1 | 22.2 | 19.8 |
| real | 54.6 | 15.3 | 40.7 | 34.0 | - | 39.2 | 36.8 |
| sketch | 54.7 | 13.6 | 34.9 | 35.7 | 43.0 | - | 36.4 |
| Avg | 43.4 | 11.2 | 27.5 | 31.3 | 36.9 | 32.3 | 30.4 |

Table 47: Individual results for each dataset pair for DomainNet64 in the SSDA setting using AdaMatch.

| SSDA DomainNet64 AdaMatch (num_target_labels=5) | | | | | | | |
|---|---|---|---|---|---|---|---|
| | clipart | infograph | painting | quickdraw | real | sketch | Avg |
| clipart | - | 12.4 | 30.4 | 47.8 | 45.8 | 42.1 | 35.7 |
| infograph | 40.4 | - | 24.1 | 39.4 | 38.5 | 29.7 | 34.4 |
| painting | 48.5 | 12.9 | - | 41.4 | 48.5 | 39.6 | 38.2 |
| quickdraw | 46.8 | 5.4 | 21.2 | - | 37.0 | 30.4 | 28.2 |
| real | 58.5 | 15.8 | 42.4 | 49.1 | - | 43.3 | 41.8 |
| sketch | 58.1 | 13.5 | 36.9 | 46.6 | 47.2 | - | 40.5 |
| Avg | 50.5 | 12.0 | 31.0 | 44.9 | 43.4 | 37.0 | 36.5 |

Table 48: Individual results for each dataset pair for DomainNet64 in the SSDA setting using AdaMatch.

| SSDA DomainNet64 AdaMatch (num_target_labels=10) | | | | | | | |
|---|---|---|---|---|---|---|---|
| | clipart | infograph | painting | quickdraw | real | sketch | Avg |
| clipart | - | 13.9 | 34.4 | 52.9 | 49.7 | 45.3 | 39.2 |
| infograph | 45.9 | - | 28.3 | 47.6 | 43.3 | 35.8 | 40.2 |
| painting | 52.8 | 14.3 | - | 49.0 | 52.0 | 43.2 | 42.3 |
| quickdraw | 52.0 | 7.2 | 27.3 | - | 44.3 | 36.8 | 33.5 |
| real | 61.2 | 17.5 | 44.7 | 54.9 | - | 46.5 | 45.0 |
| sketch | 60.1 | 15.1 | 39.4 | 52.8 | 50.7 | - | 43.6 |
| Avg | 54.4 | 13.6 | 34.8 | 51.4 | 48.0 | 41.5 | 40.6 |

Table 49: Individual results for each dataset pair for DomainNet64 in the SSDA setting using AdaMatch.

| SSDA DomainNet64 AdaMatch (num_target_labels=25) | | | | | | | |
|---|---|---|---|---|---|---|---|
| | clipart | infograph | painting | quickdraw | real | sketch | Avg |
| clipart | - | 16.8 | 40.7 | 59.0 | 56.5 | 50.3 | 44.7 |
| infograph | 54.5 | - | 35.1 | 55.4 | 51.7 | 42.7 | 47.9 |
| painting | 58.6 | 16.3 | - | 56.5 | 57.8 | 48.7 | 47.6 |
| quickdraw | 58.1 | 12.0 | 37.0 | - | 53.5 | 44.3 | 41.0 |
| real | 65.5 | 19.5 | 47.9 | 60.7 | - | 51.2 | 49.0 |
| sketch | 64.0 | 17.6 | 43.8 | 58.9 | 56.6 | - | 48.2 |
| Avg | 60.1 | 16.4 | 40.9 | 58.1 | 55.2 | 47.4 | 46.4 |

Table 50: Individual results for each dataset pair for DomainNet64 in the SSDA setting using AdaMatch.

| SSDA DomainNet64 BaselineBN (num_target_labels=1) | | | | | | | |
|---|---|---|---|---|---|---|---|
| | clipart | infograph | painting | quickdraw | real | sketch | Avg |
| clipart | - | 9.3 | 19.7 | 22.2 | 31.1 | 31.1 | 22.7 |
| infograph | 18.1 | - | 11.8 | 11.4 | 17.4 | 14.1 | 14.6 |
| painting | 31.8 | 9.7 | - | 15.8 | 37.7 | 25.1 | 24.0 |
| quickdraw | 22.3 | 2.0 | 5.6 | - | 13.0 | 13.2 | 11.2 |
| real | 42.4 | 12.6 | 34.5 | 19.2 | - | 29.2 | 27.6 |
| sketch | 43.1 | 9.7 | 22.1 | 19.3 | 29.3 | - | 24.7 |
| Avg | 31.5 | 8.7 | 18.7 | 17.6 | 25.7 | 22.5 | 20.8 |

Table 51: Individual results for each dataset pair for DomainNet64 in the SSDA setting using BaselineBN.

| SSDA DomainNet64 BaselineBN (num_target_labels=5) | | | | | | | |
|---|---|---|---|---|---|---|---|
| | clipart | infograph | painting | quickdraw | real | sketch | Avg |
| clipart | - | 10.6 | 23.7 | 37.3 | 37.6 | 35.9 | 29.0 |
| infograph | 30.2 | - | 16.2 | 29.0 | 27.1 | 21.7 | 24.8 |
| painting | 41.5 | 10.6 | - | 33.5 | 42.5 | 32.6 | 32.1 |
| quickdraw | 34.5 | 4.2 | 12.4 | - | 23.9 | 21.6 | 19.3 |
| real | 50.1 | 13.4 | 36.9 | 35.6 | - | 35.5 | 34.3 |
| sketch | 49.9 | 10.5 | 26.4 | 36.1 | 36.4 | - | 31.9 |
| Avg | 41.2 | 9.9 | 23.1 | 34.3 | 33.5 | 29.5 | 28.6 |

Table 52: Individual results for each dataset pair for DomainNet64 in the SSDA setting using BaselineBN.

| SSDA DomainNet64 BaselineBN (num_target_labels=10) | | | | | | | |
|---|---|---|---|---|---|---|---|
| | clipart | infograph | painting | quickdraw | real | sketch | Avg |
| clipart | - | 12.3 | 27.6 | 45.4 | 43.2 | 40.6 | 33.8 |
| infograph | 39.0 | - | 20.1 | 39.6 | 34.1 | 28.3 | 32.2 |
| painting | 48.0 | 12.2 | - | 42.4 | 46.9 | 37.3 | 37.4 |
| quickdraw | 42.4 | 6.2 | 17.6 | - | 32.6 | 27.9 | 25.3 |
| real | 54.4 | 15.6 | 39.1 | 44.5 | - | 40.1 | 38.7 |
| sketch | 54.5 | 12.5 | 30.1 | 44.1 | 42.3 | - | 36.7 |
| Avg | 47.7 | 11.8 | 26.9 | 43.2 | 39.8 | 34.8 | 34.0 |

Table 53: Individual results for each dataset pair for DomainNet64 in the SSDA setting using BaselineBN.

| SSDA DomainNet64 BaselineBN (num_target_labels=25) | | | | | | | |
|---|---|---|---|---|---|---|---|
| | clipart | infograph | painting | quickdraw | real | sketch | Avg |
| clipart | - | 15.1 | 36.3 | 54.4 | 51.1 | 46.7 | 40.7 |
| infograph | 51.9 | - | 30.6 | 50.2 | 45.5 | 38.4 | 43.3 |
| painting | 56.6 | 15.0 | - | 52.9 | 53.8 | 45.0 | 44.7 |
| quickdraw | 52.2 | 10.3 | 29.2 | - | 44.5 | 38.6 | 35.0 |
| real | 61.5 | 17.8 | 44.0 | 54.1 | - | 47.1 | 44.9 |
| sketch | 60.5 | 15.7 | 38.2 | 53.9 | 51.0 | - | 43.9 |
| Avg | 56.5 | 14.8 | 35.7 | 53.1 | 49.2 | 43.2 | 42.1 |

Table 54: Individual results for each dataset pair for DomainNet64 in the SSDA setting using BaselineBN.

| SSDA DomainNet64 FixMatch+ (num_target_labels=1) | | | | | | | |
|---|---|---|---|---|---|---|---|
| | clipart | infograph | painting | quickdraw | real | sketch | Avg |
| clipart | - | 11.0 | 24.5 | 29.4 | 36.7 | 36.6 | 27.6 |
| infograph | 22.4 | - | 13.5 | 14.1 | 21.7 | 17.1 | 17.8 |
| painting | 37.6 | 10.7 | - | 20.4 | 40.8 | 29.1 | 27.7 |
| quickdraw | 31.9 | 2.2 | 7.7 | - | 16.6 | 16.2 | 14.9 |
| real | 47.5 | 13.3 | 37.0 | 24.4 | - | 32.9 | 31.0 |
| sketch | 49.8 | 10.7 | 27.3 | 25.9 | 34.2 | - | 29.6 |
| Avg | 37.8 | 9.6 | 22.0 | 22.8 | 30.0 | 26.4 | 24.8 |

Table 55: Individual results for each dataset pair for DomainNet64 in the SSDA setting using FixMatch+.

| SSDA DomainNet64 FixMatch+ (num_target_labels=5) | | | | | | | |
|---|---|---|---|---|---|---|---|
| | clipart | infograph | painting | quickdraw | real | sketch | Avg |
| clipart | - | 12.3 | 29.9 | 46.5 | 44.8 | 41.4 | 35.0 |
| infograph | 39.0 | - | 19.2 | 37.7 | 32.2 | 26.6 | 30.9 |
| painting | 48.6 | 11.7 | - | 42.0 | 45.9 | 36.6 | 37.0 |
| quickdraw | 46.5 | 5.2 | 18.0 | - | 32.6 | 27.9 | 26.0 |
| real | 56.0 | 14.5 | 39.8 | 45.0 | - | 39.9 | 39.0 |
| sketch | 55.9 | 12.0 | 32.2 | 45.0 | 42.3 | - | 37.5 |
| Avg | 49.2 | 11.1 | 27.8 | 43.2 | 39.6 | 34.5 | 34.2 |

Table 56: Individual results for each dataset pair for DomainNet64 in the SSDA setting using FixMatch+.

| SSDA DomainNet64 FixMatch+ (num_target_labels=10) | | | | | | | |
|---|---|---|---|---|---|---|---|
| | clipart | infograph | painting | quickdraw | real | sketch | Avg |
| clipart | - | 13.9 | 34.9 | 53.3 | 49.9 | 45.3 | 39.5 |
| infograph | 48.5 | - | 25.1 | 47.3 | 39.7 | 33.1 | 38.7 |
| painting | 54.8 | 13.5 | - | 50.3 | 50.4 | 41.5 | 42.1 |
| quickdraw | 52.4 | 7.3 | 24.9 | - | 41.8 | 35.3 | 32.3 |
| real | 59.5 | 16.4 | 42.4 | 52.5 | - | 44.5 | 43.1 |
| sketch | 59.5 | 13.9 | 36.2 | 51.7 | 48.2 | - | 41.9 |
| Avg | 54.9 | 13.0 | 32.7 | 51.0 | 46.0 | 39.9 | 39.6 |

Table 57: Individual results for each dataset pair for DomainNet64 in the SSDA setting using FixMatch+.

| SSDA DomainNet64 FixMatch+ (num_target_labels=25) | | | | | | | |
|---|---|---|---|---|---|---|---|
| | clipart | infograph | painting | quickdraw | real | sketch | Avg |
| clipart | - | 16.8 | 42.0 | 58.9 | 56.5 | 50.2 | 44.9 |
| infograph | 57.2 | - | 35.4 | 56.0 | 49.9 | 42.6 | 48.2 |
| painting | 60.9 | 16.0 | - | 57.2 | 56.2 | 48.0 | 47.7 |
| quickdraw | 58.9 | 11.8 | 36.8 | - | 52.0 | 44.2 | 40.7 |
| real | 64.9 | 18.8 | 47.2 | 58.6 | - | 50.1 | 47.9 |
| sketch | 63.6 | 17.0 | 42.9 | 58.3 | 55.1 | - | 47.4 |
| Avg | 61.1 | 16.1 | 40.9 | 57.8 | 53.9 | 47.0 | 46.1 |

Table 58: Individual results for each dataset pair for DomainNet64 in the SSDA setting using FixMatch+.

| SSDA DomainNet64 MCD (num_target_labels=1) | | | | | | | |
|---|---|---|---|---|---|---|---|
| | clipart | infograph | painting | quickdraw | real | sketch | Avg |
| clipart | - | 6.0 | 12.3 | 15.0 | 23.4 | 21.0 | 15.5 |
| infograph | 10.2 | - | 7.2 | 7.0 | 11.2 | 7.1 | 8.5 |
| painting | 22.8 | 6.3 | - | 8.3 | 28.7 | 17.1 | 16.6 |
| quickdraw | 9.5 | 1.1 | 2.1 | - | 5.1 | 4.9 | 4.5 |
| real | 35.4 | 9.2 | 26.4 | 9.8 | - | 19.5 | 20.1 |
| sketch | 31.9 | 6.5 | 15.2 | 13.5 | 21.5 | - | 17.7 |
| Avg | 22.0 | 5.8 | 12.6 | 10.7 | 18.0 | 13.9 | 13.8 |

Table 59: Individual results for each dataset pair for DomainNet64 in the SSDA setting using MCD.

| SSDA DomainNet64 MCD (num_target_labels=5) | | | | | | | |
|---|---|---|---|---|---|---|---|
| | clipart | infograph | painting | quickdraw | real | sketch | Avg |
| clipart | - | 6.5 | 14.7 | 28.7 | 28.4 | 25.6 | 20.8 |
| infograph | 17.4 | - | 8.5 | 19.9 | 16.1 | 11.9 | 14.8 |
| painting | 28.5 | 7.2 | - | 23.1 | 31.8 | 22.3 | 22.6 |
| quickdraw | 20.1 | 2.2 | 5.1 | - | 12.5 | 12.1 | 10.4 |
| real | 39.7 | 9.9 | 27.1 | 24.1 | - | 24.8 | 25.1 |
| sketch | 37.4 | 7.0 | 17.5 | 27.2 | 26.9 | - | 23.2 |
| Avg | 28.6 | 6.6 | 14.6 | 24.6 | 23.1 | 19.3 | 19.5 |

Table 60: Individual results for each dataset pair for DomainNet64 in the SSDA setting using MCD.

| SSDA DomainNet64 MCD (num_target_labels=10) | | | | | | | |
|---|---|---|---|---|---|---|---|
| | clipart | infograph | painting | quickdraw | real | sketch | Avg |
| clipart | - | 8.2 | 18.6 | 36.5 | 33.5 | 31.3 | 25.6 |
| infograph | 24.2 | - | 11.7 | 29.4 | 22.6 | 17.4 | 21.1 |
| painting | 34.2 | 8.0 | - | 32.7 | 0 | 27.9 | 20.6 |
| quickdraw | 27.8 | 3.4 | 8.9 | - | 20.5 | 18.6 | 15.8 |
| real | 43.4 | 11.1 | 29.6 | 33.7 | - | 30.4 | 29.6 |
| sketch | 42.9 | 8.5 | 20.6 | 35.7 | 32.3 | - | 28.0 |
| Avg | 34.5 | 7.8 | 17.9 | 33.6 | 21.8 | 25.1 | 23.5 |

Table 61: Individual results for each dataset pair for DomainNet64 in the SSDA setting using MCD.

| SSDA DomainNet64 MCD (num_target_labels=25) | | | | | | | |
|---|---|---|---|---|---|---|---|
| | clipart | infograph | painting | quickdraw | real | sketch | Avg |
| clipart | - | 11.4 | 27.0 | 46.8 | 42.6 | 39.4 | 33.4 |
| infograph | 37.1 | - | 19.7 | 41.8 | 33.8 | 29.4 | 32.4 |
| painting | 44.9 | 10.5 | - | 44.2 | 43.3 | 36.8 | 35.9 |
| quickdraw | 41.1 | 6.3 | 18.1 | - | 32.4 | 29.7 | 25.5 |
| real | 49.8 | 13.4 | 34.0 | 45.2 | - | 38.7 | 36.2 |
| sketch | 49.7 | 11.6 | 29.2 | 46.0 | 41.1 | - | 35.5 |
| Avg | 44.5 | 10.6 | 25.6 | 44.8 | 38.6 | 34.8 | 33.2 |

Table 62: Individual results for each dataset pair for DomainNet64 in the SSDA setting using MCD.

| SSDA DomainNet64 NoisyStudent (num_target_labels=1) | | | | | | | |
|---|---|---|---|---|---|---|---|
| | clipart | infograph | painting | quickdraw | real | sketch | Avg |
| clipart | - | 10.5 | 23.5 | 22.5 | 35.8 | 36.6 | 25.8 |
| infograph | 23.9 | - | 16.2 | 6.7 | 22.9 | 19.1 | 17.8 |
| painting | 37.5 | 10.8 | - | 8.8 | 41.8 | 29.8 | 25.7 |
| quickdraw | 23.3 | 1.5 | 2.6 | - | 9.5 | 10.9 | 9.6 |
| real | 45.2 | 12.3 | 36.3 | 12.6 | - | 32.0 | 27.7 |
| sketch | 51.4 | 11.5 | 28.5 | 23.3 | 36.3 | - | 30.2 |
| Avg | 36.3 | 9.3 | 21.4 | 14.8 | 29.3 | 25.7 | 22.8 |

Table 63: Individual results for each dataset pair for DomainNet64 in the SSDA setting using NoisyStudent.

| SSDA DomainNet64 NoisyStudent (num_target_labels=5) | | | | | | | |
|---|---|---|---|---|---|---|---|
| | clipart | infograph | painting | quickdraw | real | sketch | Avg |
| clipart | - | 11.7 | 26.8 | 33.5 | 40.0 | 39.8 | 30.4 |
| infograph | 33.3 | - | 20.0 | 19.6 | 30.3 | 24.8 | 25.6 |
| painting | 44.5 | 11.7 | - | 21.6 | 45.8 | 34.0 | 31.5 |
| quickdraw | 31.9 | 3.1 | 5.6 | - | 16.5 | 18.9 | 15.2 |
| real | 50.8 | 13.4 | 39.1 | 30.7 | - | 37.2 | 34.2 |
| sketch | 54.6 | 12.2 | 31.1 | 33.5 | 41.5 | - | 34.6 |
| Avg | 43.0 | 10.4 | 24.5 | 27.8 | 34.8 | 30.9 | 28.6 |

Table 64: Individual results for each dataset pair for DomainNet64 in the SSDA setting using NoisyStudent.

| SSDA DomainNet64 NoisyStudent (num_target_labels=10) | | | | | | | |
|---|---|---|---|---|---|---|---|
| | clipart | infograph | painting | quickdraw | real | sketch | Avg |
| clipart | - | 12.9 | 29.6 | 42.6 | 43.8 | 43.0 | 34.4 |
| infograph | 40.6 | - | 25.4 | 33.3 | 36.1 | 30.6 | 33.2 |
| painting | 49.2 | 13.3 | - | 32.4 | 48.9 | 38.6 | 36.5 |
| quickdraw | 38.8 | 5.1 | 13.3 | - | 22.3 | 24.8 | 20.9 |
| real | 55.3 | 15.2 | 40.7 | 39.4 | - | 40.8 | 38.3 |
| sketch | 57.9 | 14.0 | 34.6 | 40.8 | 45.4 | - | 38.5 |
| Avg | 48.4 | 12.1 | 28.7 | 37.7 | 39.3 | 35.6 | 33.6 |

Table 65: Individual results for each dataset pair for DomainNet64 in the SSDA setting using NoisyStudent.

| SSDA DomainNet64 NoisyStudent (num_target_labels=25) | | | | | | | |
|---|---|---|---|---|---|---|---|
| | clipart | infograph | painting | quickdraw | real | sketch | Avg |
| clipart | - | 16.1 | 37.1 | 52.3 | 50.3 | 48.2 | 40.8 |
| infograph | 51.4 | - | 33.9 | 48.0 | 45.8 | 39.4 | 43.7 |
| painting | 56.7 | 16.0 | - | 46.7 | 54.6 | 44.8 | 43.8 |
| quickdraw | 48.7 | 9.4 | 21.6 | - | 38.0 | 37.4 | 31.0 |
| real | 60.2 | 17.9 | 45.4 | 51.0 | - | 47.1 | 44.3 |
| sketch | 61.8 | 17.3 | 41.6 | 51.6 | 51.7 | - | 44.8 |
| Avg | 55.8 | 15.3 | 35.9 | 49.9 | 48.1 | 43.4 | 41.4 |

Table 66: Individual results for each dataset pair for DomainNet64 in the SSDA setting using NoisyStudent.

| SSL DigitFive AdaMatch (num_target_labels=1) | |
|---|---|
| | Accuracy |
| mnist | 97.8 |
| mnistm | 10.1 |
| svhn | 39.8 |
| syndigit | 99.2 |
| usps | 96.3 |
| Avg | 68.6 |

Table 67: Individual results for each dataset pair for DigitFive in the SSL setting using AdaMatch.

| SSL DigitFive AdaMatch (num_target_labels=5) | |
|---|---|
| | Accuracy |
| mnist | 98.0 |
| mnistm | 97.3 |
| svhn | 96.3 |
| syndigit | 99.1 |
| usps | 95.5 |
| Avg | 97.2 |

Table 68: Individual results for each dataset pair for DigitFive in the SSL setting using AdaMatch.

| SSL DigitFive AdaMatch (num_target_labels=10) | |
|---|---|
| | Accuracy |
| mnist | 99.2 |
| mnistm | 97.1 |
| svhn | 95.7 |
| syndigit | 99.4 |
| usps | 96.5 |
| Avg | 97.6 |

Table 69: Individual results for each dataset pair for DigitFive in the SSL setting using AdaMatch.

| SSL DigitFive AdaMatch (num_target_labels=25) | |
|---|---|
| | Accuracy |
| mnist | 99.2 |
| mnistm | 98.8 |
| svhn | 95.2 |
| syndigit | 97.8 |
| usps | 97.3 |
| Avg | 97.7 |

Table 70: Individual results for each dataset pair for DigitFive in the SSL setting using AdaMatch.

| SSL DigitFive BaselineBN (num_target_labels=1) | |
|---|---|
| | Accuracy |
| mnist | 64.4 |
| mnistm | 14.6 |
| svhn | 16.9 |
| syndigit | 41.0 |
| usps | 63.0 |
| Avg | 40.0 |

Table 71: Individual results for each dataset pair for DigitFive in the SSL setting using BaselineBN.

| SSL DigitFive BaselineBN (num_target_labels=5) | |
|---|---|
| | Accuracy |
| mnist | 83.2 |
| mnistm | 29.9 |
| svhn | 36.5 |
| syndigit | 77.9 |
| usps | 81.9 |
| Avg | 61.9 |

Table 72: Individual results for each dataset pair for DigitFive in the SSL setting using BaselineBN.

| SSL DigitFive BaselineBN (num_target_labels=10) | |
|---|---|
| | Accuracy |
| mnist | 91.1 |
| mnistm | 51.5 |
| svhn | 54.0 |
| syndigit | 83.5 |
| usps | 88.5 |
| Avg | 73.7 |

Table 73: Individual results for each dataset pair for DigitFive in the SSL setting using BaselineBN.

| SSL DigitFive BaselineBN (num_target_labels=25) | |
|---|---|
| | Accuracy |
| mnist | 95.3 |
| mnistm | 78.3 |
| svhn | 69.5 |
| syndigit | 94.4 |
| usps | 94.3 |
| Avg | 86.4 |

Table 74: Individual results for each dataset pair for DigitFive in the SSL setting using BaselineBN.

**SSL DigitFive FixMatch+ (num_target_labels=1)**

|          | Accuracy |
| -------- | -------- |
| mnist    | 97.8     |
| mnistm   | 9.9      |
| svhn     | 31.5     |
| syndigit | 37.9     |
| usps     | 80.4     |
| Avg      | 51.5     |

Table 75: Individual results for each dataset pair for DigitFive in the SSL setting using FixMatch+.

**SSL DigitFive FixMatch+ (num_target_labels=5)**

|          | Accuracy |
| -------- | -------- |
| mnist    | 97.9     |
| mnistm   | 97.4     |
| svhn     | 82.2     |
| syndigit | 99.1     |
| usps     | 97.2     |
| Avg      | 94.8     |

Table 76: Individual results for each dataset pair for DigitFive in the SSL setting using FixMatch+.

**SSL DigitFive FixMatch+ (num_target_labels=10)**

|          | Accuracy |
| -------- | -------- |
| mnist    | 99.3     |
| mnistm   | 98.9     |
| svhn     | 96.8     |
| syndigit | 99.5     |
| usps     | 97.0     |
| Avg      | 98.3     |

Table 77: Individual results for each dataset pair for DigitFive in the SSL setting using FixMatch+.

**SSL DigitFive FixMatch+ (num_target_labels=25)**

|          | Accuracy |
| -------- | -------- |
| mnist    | 99.2     |
| mnistm   | 98.8     |
| svhn     | 95.9     |
| syndigit | 99.4     |
| usps     | 97.2     |
| Avg      | 98.1     |

Table 78: Individual results for each dataset pair for DigitFive in the SSL setting using FixMatch+.

| SSL DigitFive MCD (num_target_labels=1) | Accuracy |
|---|---|
| mnist | 61.6 |
| mnistm | 10.4 |
| svhn | 8.5 |
| syndigit | 9.7 |
| usps | 46.8 |
| Avg | 27.4 |

Table 79: Individual results for each dataset pair for DigitFive in the SSL setting using MCD.

| SSL DigitFive MCD (num_target_labels=5) | Accuracy |
|---|---|
| mnist | 90.9 |
| mnistm | 18.8 |
| svhn | 11.5 |
| syndigit | 40.5 |
| usps | 92.5 |
| Avg | 50.8 |

Table 80: Individual results for each dataset pair for DigitFive in the SSL setting using MCD.

| SSL DigitFive MCD (num_target_labels=10) | Accuracy |
|---|---|
| mnist | 94.8 |
| mnistm | 80.4 |
| svhn | 31.4 |
| syndigit | 92.4 |
| usps | 92.8 |
| Avg | 78.4 |

Table 81: Individual results for each dataset pair for DigitFive in the SSL setting using MCD.

| SSL DigitFive MCD (num_target_labels=25) | Accuracy |
|---|---|
| mnist | 97.1 |
| mnistm | 94.4 |
| svhn | 82.3 |
| syndigit | 96.6 |
| usps | 94.6 |
| Avg | 93.0 |

Table 82: Individual results for each dataset pair for DigitFive in the SSL setting using MCD.

**SSL DigitFive NoisyStudent (num_target_labels=1)**

|  | Accuracy |
|---|---|
| mnist | 71.2 |
| mnistm | 13.1 |
| svhn | 16.2 |
| syndigit | 40.8 |
| usps | 75.0 |
| Avg | 43.3 |

Table 83: Individual results for each dataset pair for DigitFive in the SSL setting using NoisyStudent.

**SSL DigitFive NoisyStudent (num_target_labels=5)**

|  | Accuracy |
|---|---|
| mnist | 91.0 |
| mnistm | 29.6 |
| svhn | 41.9 |
| syndigit | 84.5 |
| usps | 89.7 |
| Avg | 67.3 |

Table 84: Individual results for each dataset pair for DigitFive in the SSL setting using NoisyStudent.

**SSL DigitFive NoisyStudent (num_target_labels=10)**

|  | Accuracy |
|---|---|
| mnist | 96.8 |
| mnistm | 59.9 |
| svhn | 62.0 |
| syndigit | 90.9 |
| usps | 92.9 |
| Avg | 80.5 |

Table 85: Individual results for each dataset pair for DigitFive in the SSL setting using NoisyStudent.

**SSL DigitFive NoisyStudent (num_target_labels=25)**

|  | Accuracy |
|---|---|
| mnist | 97.9 |
| mnistm | 90.5 |
| svhn | 80.0 |
| syndigit | 95.9 |
| usps | 96.3 |
| Avg | 92.1 |

Table 86: Individual results for each dataset pair for DigitFive in the SSL setting using NoisyStudent.

| SSL DomainNet224 AdaMatch (num_target_labels=1) | |
|---|---|
| | Accuracy |
| clipart | 10.5 |
| infograph | 1.5 |
| painting | 3.9 |
| quickdraw | 18.7 |
| real | 8.3 |
| sketch | 6.5 |
| Avg | 8.2 |

Table 87: Individual results for each dataset pair for DomainNet224 in the SSL setting using AdaMatch.

| SSL DomainNet224 AdaMatch (num_target_labels=5) | |
|---|---|
| | Accuracy |
| clipart | 41.5 |
| infograph | 4.0 |
| painting | 13.0 |
| quickdraw | 42.0 |
| real | 28.6 |
| sketch | 25.1 |
| Avg | 25.7 |

Table 88: Individual results for each dataset pair for DomainNet224 in the SSL setting using AdaMatch.

| SSL DomainNet224 AdaMatch (num_target_labels=10) | |
|---|---|
| | Accuracy |
| clipart | 51.8 |
| infograph | 5.8 |
| painting | 21.7 |
| quickdraw | 51.5 |
| real | 42.4 |
| sketch | 36.5 |
| Avg | 35.0 |

Table 89: Individual results for each dataset pair for DomainNet224 in the SSL setting using AdaMatch.

| SSL DomainNet224 AdaMatch (num_target_labels=25) | |
|---|---|
| | Accuracy |
| clipart | 62.6 |
| infograph | 10.3 |
| painting | 39.5 |
| quickdraw | 59.8 |
| real | 56.5 |
| sketch | 49.6 |
| Avg | 46.4 |

Table 90: Individual results for each dataset pair for DomainNet224 in the SSL setting using AdaMatch.

| SSL DomainNet224 BaselineBN (num_target_labels=1) | |
|---|---|
| | Accuracy |
| clipart | 7.4 |
| infograph | 1.3 |
| painting | 3.4 |
| quickdraw | 14.5 |
| real | 7.0 |
| sketch | 5.3 |
| Avg | 6.5 |

Table 91: Individual results for each dataset pair for DomainNet224 in the SSL setting using BaselineBN.

| SSL DomainNet224 BaselineBN (num_target_labels=5) | |
|---|---|
| | Accuracy |
| clipart | 26.6 |
| infograph | 3.3 |
| painting | 9.1 |
| quickdraw | 33.1 |
| real | 20.1 |
| sketch | 16.6 |
| Avg | 18.1 |

Table 92: Individual results for each dataset pair for DomainNet224 in the SSL setting using BaselineBN.

| SSL DomainNet224 BaselineBN (num_target_labels=10) | |
|---|---|
| | Accuracy |
| clipart | 37.9 |
| infograph | 5.0 |
| painting | 13.5 |
| quickdraw | 42.6 |
| real | 31.1 |
| sketch | 26.9 |
| Avg | 26.2 |

Table 93: Individual results for each dataset pair for DomainNet224 in the SSL setting using BaselineBN.

| SSL DomainNet224 BaselineBN (num_target_labels=25) | |
|---|---|
| | Accuracy |
| clipart | 53.2 |
| infograph | 8.2 |
| painting | 28.0 |
| quickdraw | 53.8 |
| real | 46.2 |
| sketch | 41.4 |
| Avg | 38.5 |

Table 94: Individual results for each dataset pair for DomainNet224 in the SSL setting using BaselineBN.

| SSL DomainNet224 FixMatch+ (num_target_labels=1) | |
| --- | --- |
| | Accuracy |
| clipart | 9.3 |
| infograph | 1.4 |
| painting | 3.6 |
| quickdraw | 17.8 |
| real | 7.6 |
| sketch | 6.1 |
| Avg | 7.6 |

Table 95: Individual results for each dataset pair for DomainNet224 in the SSL setting using FixMatch+.

| SSL DomainNet224 FixMatch+ (num_target_labels=5) | |
| --- | --- |
| | Accuracy |
| clipart | 40.2 |
| infograph | 3.5 |
| painting | 10.5 |
| quickdraw | 41.6 |
| real | 26.7 |
| sketch | 22.7 |
| Avg | 24.2 |

Table 96: Individual results for each dataset pair for DomainNet224 in the SSL setting using FixMatch+.

| SSL DomainNet224 FixMatch+ (num_target_labels=10) | |
| --- | --- |
| | Accuracy |
| clipart | 50.6 |
| infograph | 5.1 |
| painting | 15.7 |
| quickdraw | 50.7 |
| real | 38.4 |
| sketch | 33.5 |
| Avg | 32.3 |

Table 97: Individual results for each dataset pair for DomainNet224 in the SSL setting using FixMatch+.

| SSL DomainNet224 FixMatch+ (num_target_labels=25) | |
| --- | --- |
| | Accuracy |
| clipart | 60.5 |
| infograph | 8.5 |
| painting | 32.2 |
| quickdraw | 58.3 |
| real | 51.2 |
| sketch | 46.5 |
| Avg | 42.9 |

Table 98: Individual results for each dataset pair for DomainNet224 in the SSL setting using FixMatch+.

**SSL DomainNet224 MCD (num_target_labels=1)**

|           | Accuracy |
|-----------|----------|
| clipart   | 2.3      |
| infograph | 0.9      |
| painting  | 1.7      |
| quickdraw | 7.5      |
| real      | 3.0      |
| sketch    | 1.2      |
| Avg       | 2.8      |

Table 99: Individual results for each dataset pair for DomainNet224 in the SSL setting using MCD.

**SSL DomainNet224 MCD (num_target_labels=5)**

|           | Accuracy |
|-----------|----------|
| clipart   | 9.2      |
| infograph | 1.9      |
| painting  | 3.8      |
| quickdraw | 25.1     |
| real      | 9.3      |
| sketch    | 4.0      |
| Avg       | 8.9      |

Table 100: Individual results for each dataset pair for DomainNet224 in the SSL setting using MCD.

**SSL DomainNet224 MCD (num_target_labels=10)**

|           | Accuracy |
|-----------|----------|
| clipart   | 16.4     |
| infograph | 3.4      |
| painting  | 7.1      |
| quickdraw | 35.7     |
| real      | 16.2     |
| sketch    | 10.6     |
| Avg       | 14.9     |

Table 101: Individual results for each dataset pair for DomainNet224 in the SSL setting using MCD.

**SSL DomainNet224 MCD (num_target_labels=25)**

|           | Accuracy |
|-----------|----------|
| clipart   | 31.8     |
| infograph | 5.7      |
| painting  | 15.1     |
| quickdraw | 47.4     |
| real      | 28.8     |
| sketch    | 27.2     |
| Avg       | 26.0     |

Table 102: Individual results for each dataset pair for DomainNet224 in the SSL setting using MCD.

**SSL DomainNet224 NoisyStudent (num_target_labels=1)**

|  | Accuracy |
|---|---|
| clipart | 9.8 |
| infograph | 1.5 |
| painting | 4.0 |
| quickdraw | 17.5 |
| real | 8.6 |
| sketch | 7.1 |
| Avg | 8.1 |

Table 103: Individual results for each dataset pair for DomainNet224 in the SSL setting using NoisyStudent.

**SSL DomainNet224 NoisyStudent (num_target_labels=5)**

|  | Accuracy |
|---|---|
| clipart | 36.4 |
| infograph | 3.9 |
| painting | 13.6 |
| quickdraw | 39.8 |
| real | 27.8 |
| sketch | 24.0 |
| Avg | 24.2 |

Table 104: Individual results for each dataset pair for DomainNet224 in the SSL setting using NoisyStudent.

**SSL DomainNet224 NoisyStudent (num_target_labels=10)**

|  | Accuracy |
|---|---|
| clipart | 48.3 |
| infograph | 5.6 |
| painting | 20.9 |
| quickdraw | 49.6 |
| real | 41.5 |
| sketch | 35.4 |
| Avg | 33.6 |

Table 105: Individual results for each dataset pair for DomainNet224 in the SSL setting using NoisyStudent.

**SSL DomainNet224 NoisyStudent (num_target_labels=25)**

|  | Accuracy |
|---|---|
| clipart | 60.8 |
| infograph | 9.7 |
| painting | 38.8 |
| quickdraw | 59.4 |
| real | 55.2 |
| sketch | 48.5 |
| Avg | 45.4 |

Table 106: Individual results for each dataset pair for DomainNet224 in the SSL setting using NoisyStudent.

| SSL DomainNet64 AdaMatch (num_target_labels=1) | |
|---|---|
| | Accuracy |
| clipart | 10.2 |
| infograph | 1.5 |
| painting | 4.0 |
| quickdraw | 18.9 |
| real | 8.5 |
| sketch | 5.6 |
| Avg | 8.1 |

Table 107: Individual results for each dataset pair for DomainNet64 in the SSL setting using AdaMatch.

| SSL DomainNet64 AdaMatch (num_target_labels=5) | |
|---|---|
| | Accuracy |
| clipart | 36.5 |
| infograph | 3.6 |
| painting | 13.3 |
| quickdraw | 41.7 |
| real | 26.9 |
| sketch | 21.0 |
| Avg | 23.8 |

Table 108: Individual results for each dataset pair for DomainNet64 in the SSL setting using AdaMatch.

| SSL DomainNet64 AdaMatch (num_target_labels=10) | |
|---|---|
| | Accuracy |
| clipart | 46.4 |
| infograph | 5.8 |
| painting | 20.8 |
| quickdraw | 49.9 |
| real | 37.9 |
| sketch | 29.6 |
| Avg | 31.7 |

Table 109: Individual results for each dataset pair for DomainNet64 in the SSL setting using AdaMatch.

| SSL DomainNet64 AdaMatch (num_target_labels=25) | |
|---|---|
| | Accuracy |
| clipart | 56.2 |
| infograph | 9.3 |
| painting | 33.3 |
| quickdraw | 57.7 |
| real | 50.3 |
| sketch | 41.4 |
| Avg | 41.4 |

Table 110: Individual results for each dataset pair for DomainNet64 in the SSL setting using AdaMatch.

| SSL DomainNet64 BaselineBN (num_target_labels=1) | |
|---|---|
| | Accuracy |
| clipart | 6.9 |
| infograph | 1.4 |
| painting | 3.1 |
| quickdraw | 13.2 |
| real | 6.6 |
| sketch | 4.3 |
| Avg | 5.9 |

Table 111: Individual results for each dataset pair for DomainNet64 in the SSL setting using BaselineBN.

| SSL DomainNet64 BaselineBN (num_target_labels=5) | |
|---|---|
| | Accuracy |
| clipart | 23.6 |
| infograph | 3.2 |
| painting | 9.1 |
| quickdraw | 31.2 |
| real | 18.2 |
| sketch | 13.9 |
| Avg | 16.5 |

Table 112: Individual results for each dataset pair for DomainNet64 in the SSL setting using BaselineBN.

| SSL DomainNet64 BaselineBN (num_target_labels=10) | |
|---|---|
| | Accuracy |
| clipart | 33.9 |
| infograph | 4.8 |
| painting | 14.2 |
| quickdraw | 40.3 |
| real | 28.2 |
| sketch | 22.1 |
| Avg | 23.9 |

Table 113: Individual results for each dataset pair for DomainNet64 in the SSL setting using BaselineBN.

| SSL DomainNet64 BaselineBN (num_target_labels=25) | |
|---|---|
| | Accuracy |
| clipart | 48.2 |
| infograph | 7.8 |
| painting | 25.7 |
| quickdraw | 51.1 |
| real | 41.5 |
| sketch | 34.9 |
| Avg | 34.9 |

Table 114: Individual results for each dataset pair for DomainNet64 in the SSL setting using BaselineBN.

| **SSL DomainNet64 FixMatch+ (num_target_labels=1)** | |
| --- | --- |
| | Accuracy |
| clipart | 9.7 |
| infograph | 1.4 |
| painting | 3.9 |
| quickdraw | 17.1 |
| real | 7.5 |
| sketch | 6.0 |
| Avg | 7.6 |

Table 115: Individual results for each dataset pair for DomainNet64 in the SSL setting using FixMatch+.

| **SSL DomainNet64 FixMatch+ (num_target_labels=5)** | |
| --- | --- |
| | Accuracy |
| clipart | 37.6 |
| infograph | 3.5 |
| painting | 13.1 |
| quickdraw | 42.3 |
| real | 27.2 |
| sketch | 21.1 |
| Avg | 24.1 |

Table 116: Individual results for each dataset pair for DomainNet64 in the SSL setting using FixMatch+.

| **SSL DomainNet64 FixMatch+ (num_target_labels=10)** | |
| --- | --- |
| | Accuracy |
| clipart | 48.1 |
| infograph | 5.3 |
| painting | 18.9 |
| quickdraw | 50.9 |
| real | 39.0 |
| sketch | 30.6 |
| Avg | 32.1 |

Table 117: Individual results for each dataset pair for DomainNet64 in the SSL setting using FixMatch+.

| **SSL DomainNet64 FixMatch+ (num_target_labels=25)** | |
| --- | --- |
| | Accuracy |
| clipart | 57.2 |
| infograph | 8.8 |
| painting | 32.7 |
| quickdraw | 58.0 |
| real | 50.5 |
| sketch | 41.6 |
| Avg | 41.5 |

Table 118: Individual results for each dataset pair for DomainNet64 in the SSL setting using FixMatch+.

| SSL DomainNet64 MCD (num_target_labels=1) | Accuracy |
|---|---|
| clipart | 3.0 |
| infograph | 1.0 |
| painting | 1.8 |
| quickdraw | 7.8 |
| real | 3.5 |
| sketch | 1.6 |
| Avg | 3.1 |

Table 119: Individual results for each dataset pair for DomainNet64 in the SSL setting using MCD.

| SSL DomainNet64 MCD (num_target_labels=5) | Accuracy |
|---|---|
| clipart | 10.5 |
| infograph | 2.3 |
| painting | 5.0 |
| quickdraw | 25.1 |
| real | 11.3 |
| sketch | 6.9 |
| Avg | 10.2 |

Table 120: Individual results for each dataset pair for DomainNet64 in the SSL setting using MCD.

| SSL DomainNet64 MCD (num_target_labels=10) | Accuracy |
|---|---|
| clipart | 19.6 |
| infograph | 3.5 |
| painting | 9.5 |
| quickdraw | 35.1 |
| real | 18.7 |
| sketch | 13.6 |
| Avg | 16.7 |

Table 121: Individual results for each dataset pair for DomainNet64 in the SSL setting using MCD.

| SSL DomainNet64 MCD (num_target_labels=25) | Accuracy |
|---|---|
| clipart | 34.7 |
| infograph | 5.6 |
| painting | 17.3 |
| quickdraw | 47.1 |
| real | 32.3 |
| sketch | 27.9 |
| Avg | 27.5 |

Table 122: Individual results for each dataset pair for DomainNet64 in the SSL setting using MCD.

| SSL DomainNet64 NoisyStudent (num_target_labels=1) | Accuracy |
|---|---|
| clipart | 8.7 |
| infograph | 1.5 |
| painting | 4.1 |
| quickdraw | 16.7 |
| real | 7.6 |
| sketch | 5.4 |
| Avg | 7.3 |

Table 123: Individual results for each dataset pair for DomainNet64 in the SSL setting using NoisyStudent.

| SSL DomainNet64 NoisyStudent (num_target_labels=5) | Accuracy |
|---|---|
| clipart | 31.8 |
| infograph | 3.5 |
| painting | 11.8 |
| quickdraw | 37.1 |
| real | 24.3 |
| sketch | 17.9 |
| Avg | 21.1 |

Table 124: Individual results for each dataset pair for DomainNet64 in the SSL setting using NoisyStudent.

| SSL DomainNet64 NoisyStudent (num_target_labels=10) | Accuracy |
|---|---|
| clipart | 42.3 |
| infograph | 5.3 |
| painting | 18.6 |
| quickdraw | 46.7 |
| real | 35.6 |
| sketch | 27.2 |
| Avg | 29.3 |

Table 125: Individual results for each dataset pair for DomainNet64 in the SSL setting using NoisyStudent.

| SSL DomainNet64 NoisyStudent (num_target_labels=25) | Accuracy |
|---|---|
| clipart | 54.6 |
| infograph | 8.7 |
| painting | 31.7 |
| quickdraw | 57.4 |
| real | 49.4 |
| sketch | 40.3 |
| Avg | 40.4 |

Table 126: Individual results for each dataset pair for DomainNet64 in the SSL setting using NoisyStudent.

| UDA DigitFive AdaMatch | | | | | |
|---|---|---|---|---|---|
| | mnist | mnistm | svhn | syndigit | usps | Avg |
| mnist | - | 99.2 | 96.9 | 99.7 | 97.8 | 98.4 |
| mnistm | 99.4 | - | 96.9 | 99.7 | 97.8 | 98.5 |
| svhn | 99.3 | 98.9 | - | 99.6 | 90.4 | 97.0 |
| syndigit | 99.4 | 99.0 | 97.0 | - | 95.8 | 97.8 |
| usps | 99.3 | 98.9 | 96.6 | 94.9 | - | 97.4 |
| Avg | 99.4 | 99.0 | 96.8 | 98.5 | 95.5 | 97.8 |

Table 127: Individual results for each dataset pair for DigitFive in the UDA setting using AdaMatch.

| UDA DigitFive BaselineBN | | | | | |
|---|---|---|---|---|---|
| | mnist | mnistm | svhn | syndigit | usps | Avg |
| mnist | - | 54.3 | 55.0 | 76.0 | 96.3 | 70.4 |
| mnistm | 96.5 | - | 52.4 | 81.8 | 88.1 | 79.7 |
| svhn | 40.4 | 32.0 | - | 86.3 | 39.9 | 49.6 |
| syndigit | 67.7 | 30.0 | 79.7 | - | 70.5 | 62.0 |
| usps | 78.6 | 27.3 | 31.9 | 64.8 | - | 50.6 |
| Avg | 70.8 | 35.9 | 54.8 | 77.2 | 73.7 | 62.5 |

Table 128: Individual results for each dataset pair for DigitFive in the UDA setting using BaselineBN.

| UDA DigitFive FixMatch+ | | | | | |
|---|---|---|---|---|---|
| | mnist | mnistm | svhn | syndigit | usps | Avg |
| mnist | - | 99.2 | 97.0 | 99.7 | 97.9 | 98.4 |
| mnistm | 99.4 | - | 97.0 | 99.6 | 85.1 | 95.3 |
| svhn | 99.2 | 98.9 | - | 99.5 | 87.3 | 96.2 |
| syndigit | 99.3 | 99.0 | 96.9 | - | 97.2 | 98.1 |
| usps | 99.3 | 99.0 | 62.6 | 99.8 | - | 90.2 |
| Avg | 99.3 | 99.0 | 88.4 | 99.7 | 91.9 | 95.6 |

Table 129: Individual results for each dataset pair for DigitFive in the UDA setting using FixMatch+.

| UDA DigitFive MCD | | | | | |
|---|---|---|---|---|---|
| | mnist | mnistm | svhn | syndigit | usps | Avg |
| mnist | - | 10.7 | 13.3 | 11.1 | 90.0 | 31.3 |
| mnistm | 92.2 | - | 13.7 | 13.2 | 52.0 | 42.8 |
| svhn | 97.0 | 53.3 | - | 97.9 | 90.4 | 84.7 |
| syndigit | 98.3 | 27.9 | 66.0 | - | 91.7 | 71.0 |
| usps | 98.1 | 10.0 | 6.4 | 9.8 | - | 31.1 |
| Avg | 96.4 | 25.5 | 24.9 | 33.0 | 81.0 | 52.1 |

Table 130: Individual results for each dataset pair for DigitFive in the UDA setting using MCD.

| UDA DigitFive NoisyStudent | | | | | | |
|---|---|---|---|---|---|---|
| | mnist | mnistm | svhn | syndigit | usps | Avg |
| mnist | - | 30.4 | 51.4 | 81.8 | 97.2 | 65.2 |
| mnistm | 99.2 | - | 63.0 | 89.0 | 97.0 | 87.0 |
| svhn | 82.8 | 45.2 | - | 98.9 | 83.9 | 77.7 |
| syndigit | 92.6 | 50.5 | 90.0 | - | 87.1 | 80.0 |
| usps | 95.5 | 25.2 | 30.0 | 75.3 | - | 56.5 |
| Avg | 92.5 | 37.8 | 58.6 | 86.3 | 91.3 | 73.3 |

Table 131: Individual results for each dataset pair for DigitFive in the UDA setting using NoisyStudent.

| UDA DomainNet224 AdaMatch | | | | | | |
|---|---|---|---|---|---|---|
| | clipart | infograph | painting | quickdraw | real | sketch | Avg |
| clipart | - | 14.8 | 35.3 | 26.8 | 46.5 | 46.5 | 34.0 |
| infograph | 21.7 | - | 11.7 | 0.3 | 20.0 | 0.2 | 10.8 |
| painting | 45.3 | 13.5 | - | 13.2 | 48.1 | 41.0 | 32.2 |
| quickdraw | 33.6 | 2.4 | 10.1 | - | 19.1 | 21.1 | 17.3 |
| real | 56.0 | 16.6 | 47.6 | 22.9 | - | 42.4 | 37.1 |
| sketch | 60.2 | 17.0 | 42.9 | 34.2 | 49.1 | - | 40.7 |
| Avg | 43.4 | 12.9 | 29.5 | 19.5 | 36.6 | 30.2 | 28.7 |

Table 132: Individual results for each dataset pair for DomainNet224 in the UDA setting using AdaMatch.

| UDA DomainNet224 BaselineBN | | | | | | |
|---|---|---|---|---|---|---|
| | clipart | infograph | painting | quickdraw | real | sketch | Avg |
| clipart | - | 11.4 | 23.4 | 12.1 | 33.4 | 34.9 | 23.0 |
| infograph | 16.2 | - | 13.8 | 0.9 | 16.5 | 11.5 | 11.8 |
| painting | 33.6 | 11.2 | - | 1.0 | 41.2 | 25.7 | 22.5 |
| quickdraw | 11.2 | 0.8 | 2.5 | - | 5.5 | 7.1 | 5.4 |
| real | 43.4 | 13.7 | 39.4 | 2.1 | - | 29.6 | 25.6 |
| sketch | 47.9 | 11.0 | 28.7 | 3.8 | 33.1 | - | 24.9 |
| Avg | 30.5 | 9.6 | 21.6 | 4.0 | 25.9 | 21.8 | 18.9 |

Table 133: Individual results for each dataset pair for DomainNet224 in the UDA setting using BaselineBN.

| UDA DomainNet224 FixMatch+ | | | | | | |
|---|---|---|---|---|---|---|
| | clipart | infograph | painting | quickdraw | real | sketch | Avg |
| clipart | - | 11.6 | 26.4 | 11.2 | 37.6 | 37.8 | 24.9 |
| infograph | 15.8 | - | 6.5 | 0.6 | 8.6 | 11.4 | 8.6 |
| painting | 39.0 | 11.6 | - | 4.2 | 42.7 | 31.4 | 25.8 |
| quickdraw | 15.8 | 1.3 | 5.0 | - | 11.5 | 12.2 | 9.2 |
| real | 48.1 | 13.8 | 39.8 | 7.3 | - | 33.4 | 28.5 |
| sketch | 50.3 | 11.8 | 30.7 | 11.8 | 34.2 | - | 27.8 |
| Avg | 33.8 | 10.0 | 21.7 | 7.0 | 26.9 | 25.2 | 20.8 |

Table 134: Individual results for each dataset pair for DomainNet224 in the UDA setting using FixMatch+.

| UDA DomainNet224 MCD | | | | | | |
|---|---|---|---|---|---|---|
| | clipart | infograph | painting | quickdraw | real | sketch | Avg |
| clipart | - | 7.7 | 15.2 | 15.2 | 25.8 | 23.4 | 17.5 |
| infograph | 9.2 | - | 7.0 | 1.6 | 11.2 | 6.5 | 7.1 |
| painting | 25.6 | 7.0 | - | 5.3 | 32.1 | 18.8 | 17.8 |
| quickdraw | 7.4 | 0.7 | 0.7 | - | 1.6 | 4.1 | 2.9 |
| real | 38.4 | 10.5 | 32.4 | 7.7 | - | 22.9 | 22.4 |
| sketch | 37.2 | 8.2 | 22.9 | 14.7 | 26.7 | - | 21.9 |
| Avg | 23.6 | 6.8 | 15.6 | 8.9 | 19.5 | 15.1 | 14.9 |

Table 135: Individual results for each dataset pair for DomainNet224 in the UDA setting using MCD.

| UDA DomainNet224 NoisyStudent | | | | | | |
|---|---|---|---|---|---|---|
| | clipart | infograph | painting | quickdraw | real | sketch | Avg |
| clipart | - | 12.9 | 28.5 | 20.4 | 39.5 | 42.3 | 28.7 |
| infograph | 24.5 | - | 18.9 | 3.1 | 23.9 | 18.3 | 17.7 |
| painting | 37.6 | 11.4 | - | 3.4 | 40.9 | 30.9 | 24.8 |
| quickdraw | 20.9 | 1.0 | 2.7 | - | 8.7 | 13.2 | 9.3 |
| real | 47.8 | 13.7 | 39.8 | 7.7 | - | 32.6 | 28.3 |
| sketch | 55.2 | 14.8 | 37.1 | 22.8 | 41.8 | - | 34.3 |
| Avg | 37.2 | 10.8 | 25.4 | 11.5 | 31.0 | 27.5 | 23.9 |

Table 136: Individual results for each dataset pair for DomainNet224 in the UDA setting using NoisyStudent.

| UDA DomainNet64 AdaMatch | | | | | | |
|---|---|---|---|---|---|---|
| | clipart | infograph | painting | quickdraw | real | sketch | Avg |
| clipart | - | 12.2 | 29.1 | 27.7 | 40.9 | 40.8 | 30.1 |
| infograph | 22.3 | - | 18.2 | 5.4 | 24.7 | 19.9 | 18.1 |
| painting | 39.7 | 11.7 | - | 12.6 | 43.4 | 34.9 | 28.5 |
| quickdraw | 24.6 | 2.4 | 8.4 | - | 16.4 | 15.5 | 13.5 |
| real | 52.2 | 15.0 | 41.6 | 13.3 | - | 38.6 | 32.1 |
| sketch | 53.4 | 13.6 | 35.6 | 25.4 | 41.5 | - | 33.9 |
| Avg | 38.4 | 11.0 | 26.6 | 16.9 | 33.4 | 29.9 | 26.0 |

Table 137: Individual results for each dataset pair for DomainNet64 in the UDA setting using AdaMatch.

| UDA DomainNet64 BaselineBN | | | | | | |
|---|---|---|---|---|---|---|
| | clipart | infograph | painting | quickdraw | real | sketch | Avg |
| clipart | - | 9.5 | 19.0 | 6.3 | 29.4 | 29.9 | 18.8 |
| infograph | 12.9 | - | 10.8 | 0.6 | 14.0 | 9.5 | 9.6 |
| painting | 27.3 | 9.3 | - | 1.0 | 36.3 | 21.8 | 19.1 |
| quickdraw | 10.3 | 0.8 | 1.9 | - | 4.8 | 5.9 | 4.7 |
| real | 39.3 | 11.9 | 34.6 | 2.2 | - | 25.5 | 22.7 |
| sketch | 40.9 | 8.5 | 20.4 | 3.3 | 26.0 | - | 19.8 |
| Avg | 26.1 | 8.0 | 17.3 | 2.7 | 22.1 | 18.5 | 15.8 |

Table 138: Individual results for each dataset pair for DomainNet64 in the UDA setting using BaselineBN.

| UDA DomainNet64 FixMatch+ | | | | | | | |
|---|---|---|---|---|---|---|---|
| | clipart | infograph | painting | quickdraw | real | sketch | Avg |
| clipart | - | 10.2 | 24.6 | 14.6 | 35.7 | 35.3 | 24.1 |
| infograph | 17.9 | - | 14.3 | 2.2 | 18.7 | 14.2 | 13.5 |
| painting | 34.7 | 10.7 | - | 4.2 | 39.7 | 27.7 | 23.4 |
| quickdraw | 23.7 | 1.4 | 5.1 | - | 11.2 | 10.2 | 10.3 |
| real | 45.4 | 13.5 | 37.4 | 6.6 | - | 30.7 | 26.7 |
| sketch | 48.1 | 10.3 | 26.6 | 10.4 | 31.5 | - | 25.4 |
| Avg | 34.0 | 9.2 | 21.6 | 7.6 | 27.4 | 23.6 | 20.6 |

Table 139: Individual results for each dataset pair for DomainNet64 in the UDA setting using FixMatch+.

| UDA DomainNet64 MCD | | | | | | | |
|---|---|---|---|---|---|---|---|
| | clipart | infograph | painting | quickdraw | real | sketch | Avg |
| clipart | - | 7.5 | 15.9 | 12.9 | 26.2 | 25.6 | 17.6 |
| infograph | 11.3 | - | 9.4 | 2.3 | 13.3 | 8.6 | 9.0 |
| painting | 25.8 | 7.5 | - | 4.3 | 32.1 | 21.4 | 18.2 |
| quickdraw | 8.0 | 0.6 | 0.6 | - | 1.6 | 4.6 | 3.1 |
| real | 37.2 | 10.1 | 29.8 | 6.8 | - | 22.9 | 21.4 |
| sketch | 37.3 | 7.8 | 20.2 | 11.6 | 25.0 | - | 20.4 |
| Avg | 23.9 | 6.7 | 15.2 | 7.6 | 19.6 | 16.6 | 14.9 |

Table 140: Individual results for each dataset pair for DomainNet64 in the UDA setting using MCD.

| UDA DomainNet64 NoisyStudent | | | | | | | |
|---|---|---|---|---|---|---|---|
| | clipart | infograph | painting | quickdraw | real | sketch | Avg |
| clipart | - | 10.9 | 24.3 | 18.9 | 35.4 | 36.8 | 25.3 |
| infograph | 21.2 | - | 16.9 | 2.6 | 21.8 | 16.0 | 15.7 |
| painting | 36.1 | 11.7 | - | 3.6 | 41.5 | 28.5 | 24.3 |
| quickdraw | 19.0 | 0.9 | 1.5 | - | 6.8 | 7.2 | 7.1 |
| real | 44.0 | 13.8 | 38.3 | 6.6 | - | 30.6 | 26.7 |
| sketch | 50.0 | 12.0 | 30.4 | 18.2 | 36.0 | - | 29.3 |
| Avg | 34.1 | 9.9 | 22.3 | 10.0 | 28.3 | 23.8 | 21.4 |

Table 141: Individual results for each dataset pair for DomainNet64 in the UDA setting using NoisyStudent.

