# OpenReview forum: "AdaMatch: A Unified Approach to Semi-Supervised Learning and Domain Adaptation"
_ICLR.cc/2022/Conference — ICLR 2022 Poster_

### Official Review · Reviewer_wiKA · 2021-10-23

**Correctness:** 3
**Technical Novelty And Significance:** 2
**Empirical Novelty And Significance:** 3
**Recommendation:** 5
**Confidence:** 4

**Main Review:**

Strengths:
- The description of the relation between the tasks (SSL, UDA, and SSDA) helps to understand the main problem.  For example, Table 1 was really useful.

- Strong results that improve over the compared methods on three setups.

- Ablation study that shows the contribution of each part of the model.

Weaknesses:
- The contributions are not clear.  Most of the steps used are taken from previous methods in the literature and adapted.  For instance, the distribution alignment comes from Berthelot et al. 2020, or the loss from FixMatch (Sohn et al. 2020).  Please, clearly state what are your contributions over the state-of-the-art.  If the contribution is the mixture of existing ideas, then say so, instead of misleading the reader.

- The diagram of Fig. 1 is hard to follow.  For instance, the upper stream should be named as the variable $Z''$ and the bottom one $Z'$ (if I understood it correctly).  Similarly, the relative threshold seems to depend only from the source, yet it depends on the source and target (6).  Also, what is the operation that produces the second distribution on the middle row?  Is it the alignment w.r.t. the source and target weak?  If so, shouldn't the arrow mix both instead of only coming from the source?

- It is not clear what you are trying to do with the random logit interpolation.

  I don't understand how the interpolation relates to rather equalizing $Z_{\text{SL}}'$ or $Z''_{\text{SL}}$.  Since it is a random variable, there is no optimization that will happen on them, but just a mixture of them.

  $Z_{\text{SL}}'$ has the information of the source and target, while $Z''_{\text{SL}}$ only the source information.  Is your interpolation mixing that information as a form of regularization?  Or is $f$ shared in a specific way that let you achieve the claim of minimizing the loss of every point between these logits? Is not only through a normalization on the batch that the logits differ?

  Moreover, since the $\lambda$ is random, you can't guarantee that you will learn a mixture of interest (i.e., an optimal $\lambda$), but instead you are making it robust to it (somehow).

  Could you explain what you are trying to achieve here?  Is my interpretation correct?

- It is not clear what the claim that the random logit interpolation will make the logits either equal or to belong to an infimum.  How can the interpolation control what the loss landscape looks like?

- It seems from (1) and (2) that you call the model twice for mixing the source and target data (1) to learn some representations, and then alone with only the source data (2).  However, after these equations you mention that the model is called twice for each augmentation type.  Which one is it?  Or are you doing both?

- If the model $f(\cdot)$ already outputs the logits for the classes, why do you compute the pseudo-labels through another round of softmax functions (5)?

- You should minimally describe the augmentations you use to give a reader an idea of what "weak" and "strong" means.  Leave the details in the reference, but at least make a setup for the reader.

**Summary Of The Paper:**

The paper extends FixMatch as a unified approach for semi-Supervised Learning, Unsupervised Domain Adaptation, and Semi-supervised Domain Adaptation.  The proposal learns representations with a cross-entropy loss between labels and logits for a strong and weak augmentation versions of the source data while the target data helps with regularization.  The proposal uses a random mixture the logits of the two augmentations, and normalizes the distribution of the pseudo-labels with the ratio of the expected logits of the augmentations.

The contribution is the mixture of aligning the label distributions by normalizing them through the ratio between the source and target predicted labels, using relative thresholds to compare source and target label distributions, and the random logit interpolation to regularize the representations.  This combinations let the method outperform the reported methods in the three tasks on two datasets.

**Summary Of The Review:**

The proposal extends FixMatch to work on three related tasks, and outperforms existing methods on two datasets.  The proposal mixes existing approaches for normalizing label distributions to train the model with a cross-entropy loss on different versions of the data to provide regularization. Moreover, the paper also proposes a random logit interpolation as means of regularization.  However, the intent of how these pieces were selected and how they work together is not clear to me.  The paper demonstrates empirically that the parts are necessary through an ablation study, yet I don't know why they are needed.

Overall, the paper presents an engineer method that works, although why escapes me.

---

> ### Author Response · Authors · 2021-11-12
> **Response to Reviewer wiKA**
>
> We thank the reviewer for their time and feedback on our paper. We respond to some of the reviewer's comments:
>
> 1. *Contributions are not clear*:
>     - Though distribution alignment and FixMatch have been introduced before, our contribution is showing that these techniques, combined with our modifications, can actually solve the new problem of distribution shift.
>     - Moreover as stated in Section 3.1, random logit interpolation, relative confidence thresholding, and our version of distribution alignment are all novel contributions that did not appear in prior work.
>     - We also contribute to the field by evaluating SSL and UDA methods across all three settings (UDA, SSL, and SSDA). Prior work typically only evaluated one style of algorithm on the setting it was designed for. Our results showcase the various strengths of the algorithms across each task, providing a valuable benchmark for the community.
>
> 2. *Figure 1 is hard to follow*:  We appreciate your constructive feedback and will update the figure and caption to improve clarity.
>     - Your understanding of Z’ and Z’’ is correct. We will include these variable names in an updated figure.
>     - The target data implicitly influences the relative confidence threshold through the Z’ path.
>     - The second distribution is produced by distribution alignment, where the target distribution comes from the blue distribution above.
>
> 3. *Interpretation of random logit interpolation*:
>     The key point is that we randomly choose the interpolation value $\lambda$, and then minimize the loss on this randomly interpolated value. Because we randomly choose the point on the line each time, over the course of training we can ensure that the entire line segment between ZSL’ and ZSL’’ reaches low loss. Another way to achieve the same result would be to divide the interval into N (a large number) of different segments, and then minimize the loss on all N points. However this would be computationally expensive and so by randomly choosing a new $\lambda$ each time we achieve the same result without increasing the computation cost by a factor of N. We will update the text to make this more clear.
>
> 4. *How can the interpolation control what the loss landscape looks like?*:
>     - The interpolation controls which points the model has a chance to minimize its loss with respect to, so it affects the landscape as the gradient of the loss flows back through the output of the interpolation.
>     - The loss could be minimized between ZSL’ and ZSL’’ if the model makes these two points equal. However, if the points are not equal, then we want to allow the model to instead minimize any point found by interpolating between these two values. We don’t force the model to choose one of these over the other, and so both are valid.
>
> 5. *Do we call the model twice?*: We generate both the weakly and strongly augmented data and then concatenate them together. Then we call the model twice on this new concatenated minibatch on both the combined source-and-target batch (eqn 1) and also the source batch (eqn 2).
>
> 6. *why do you compute the pseudo-labels through another round of softmax functions*: The output of f is the logits, not a probability distribution, and computing the softmax function is necessary because we want the pseudo labels to be a probability distribution.
>
> 7. *Which augmentations do we use*: We describe the method using the general “weak” and “strong” terms because they can be dependent on the task. For all of our results, and our implementation’s definition: Weak is shift and mirror about the horizontal axis. Strong is weak augmentation plus the addition of cutout. Thank you for this suggestion, we will add this to the paper.

---

> > ### Comment · Reviewer_wiKA · 2021-11-18
> > **Thanks for the answers**
> >
> > I understand your paper better now.  I still have my doubts regarding the claimed contributions, but I see where are you coming from.
> >
> > One last thing that is still not clear to me is the double softmax.  You claim that your logits need to be transformed into a distribution, but aren't the logits log-probabilities by definition?  So, the probabilities already come from a distribution.  What am I missing here?

---

> > > ### Author Response · Authors · 2021-11-21
> > > **Logit clarification**
> > >
> > > By logits, we refer to the vector of raw (non-normalized) predictions that the model generates, which is ordinarily then passed to a normalization function. We use the logits as input to the softmax function, which generates a vector of (normalized) probabilities with one value for each possible class. We only call the softmax function once to generate the pseudo-labels in equation (5). Does that clear up the confusion?

---

> ### Author Response · Authors · 2021-11-12
> **Response to Reviewer wiKA continued**
>
> *Response to Review Summary:* Our major motivation for introducing new components was to help address the distribution shift between the labeled and unlabeled data, which is not something SSL methods are typically designed to handle.  A takeaway from our paper is that SSL methods are able to perform well on problems with distribution shift, but they can achieve even stronger performance if we modify them in ways that explicitly address the domain differences. For each of the components we add, we describe why they are necessary to help with the new problems that come from distribution shift:
>
>    - Random logit “has the effect of producing batch statistics that are more representative of both domains" (Section 3.1: Random Logit Interpolation) and "creates an implicit constraint to align the source and target domains in logit space" (Section 3.1: Overview).
>    - Distribution alignment “helps constrain the distribution of the class predictions to be more aligned with the true distribution”  (Section 3.1: Distribution Alignment). If we left it out, then the model would have no incentive to match the target distribution to the source distribution.
>    - Relative confidence thresholding addresses the issue that “models are poorly calibrated … on out-of-distribution data." (Section 3.1: Relative Confidence Threshold) By including relative confidence thresholding, we can ensure that the target data is used as pseudo labels as often as it should be.
>
> We thank the reviewer for their constructive feedback and we will add a discussion that explains more carefully why we introduced these components and summarize how they help with distribution shift.

---

### Official Review · Reviewer_Jgga · 2021-10-31

**Correctness:** 3
**Technical Novelty And Significance:** 3
**Empirical Novelty And Significance:** 3
**Recommendation:** 6
**Confidence:** 4

**Main Review:**

The strengths of the paper:
+ The paper is well-written.
+ The paper introduces an AdaMatch method to boost accuracy on domain shift for UDA, SSL and SSDA, which extends the existing FixMatch in three aspects.
+ The AdaMatch achieves the state-of-the-art accuracy on the SSL, UDA and SSDA tasks using the same hyper-parameters regardless of the dataset.
+ Experiments evaluation is extensive, and the ablation analysis helps understand the importance of each improvement in the AdaMatch.

The weaknesses of the paper:
- In loss function (11), how does \mu(t) boost the accuracy compared to the case that \mu(t) = 1 in the entire training process.
- In section 5.1.1 UDA WITH PRE-TRAINING, why is MCD selected as a baseline for comparison?
- In Figure 4 (center), the accuracy increases when the unlabeled to labeled data ratio (uratio) is increased, is this Figure correctly plotted?

**Summary Of The Paper:**

To boost accuracy on domain shifts, this paper proposes an AdaMatch method for unsupervised domain adaptation (UDA), semi-supervised learning (SSL) and semi-supervised domain adaptation (SSDA). The proposed AdaMatch extends the existing FixMatch in three aspects, and it achieves the state-of-the-art accuracy using the same hyper-parameters regardless of the dataset or task.

**Summary Of The Review:**

The contribution is significant and somewhat new, and the experimental evaluation is extensive. However, three concerns in the weaknesses of the paper should be clarified in the paper.

---

> ### Author Response · Authors · 2021-11-12
> **Response to Reviewer Jgga**
>
> We thank the reviewer for their time and feedback on our paper. We respond to some of the reviewer's comments:
>
> 1. *Role of $\mu(t)$*: It does not change the overall accuracy, but just makes the model converge faster so we do not need to train for as long.
>
> 2. *MCD as a baseline* MCD has the highest published accuracy on single source domain adaptation for DomainNet, and so we use it as the best baseline possible to match this configuration.
>
> 3. *Accuracy increases when uratio increases:* Yes, this figure is correct. Uratio defines the ratio of unlabeled data to labeled data within a mini-batch, and not the ratio between the total number of unlabeled and labeled examples seen over the course of training.
>
> We will add clarifications for all of these points to the paper. Thank you!

---

> > ### Comment · Reviewer_Jgga · 2021-11-30
> > **Thanks for the response**
> >
> > Thanks for the response and addressing the previous concerns.

---

### Official Review · Reviewer_aiq7 · 2021-11-01

**Correctness:** 3
**Technical Novelty And Significance:** 2
**Empirical Novelty And Significance:** 4
**Recommendation:** 6
**Confidence:** 4

**Main Review:**



Strength

- It is interesting and should be practically useful to take a unified scheme for SSL and DA.

- A thorough empirical study validates the advantage of the proposed method in the setting of UDA, SSL, and SSDA.

- This paper is well-written and easy to follow.

Weakness

- The technical novelty of the proposed method is somewhat limited. AdaMatch is based on FixMatch and contains three modifications: random logit interpolation, prediction-based distribution alignment, and relative confidence threshold.

  - The random logit interpolation is somewhat novel here, but its siginificancy seems slight (Table 4).

  - The modification on distributional alignment is incremental. It is quite natural to use model's predictions for estimating class-wise weights of loss, especially in the DA setting (e.g. [R1]).

     [R1] "Partial Adversarial Domain Adaptation," ECCV 2018.

  - The modification on confidence threshold is also incremental. In addition, the idea to dynamically change the threshold for pseudo-labeling during training can be found in recent SSL method [R2].

     [R2] "Dash: Semi-Supervised Learning with Dynamic Thresholding," ICML 2021.


- I understand the intuition on why we need relative confidence threshold, but does it work properly in the early stage of the training? When the model is not sufficiently trained, c_\tau can be relatively small, which might induce accepting many incorrect pseudo-labels. Although this issue may be alleviated by adopting \mu in Eq. (11), it would be worth discussing somewhere in the manuscript.


Minor concerns that do not affect my score

- It should be interesting to see exclude-one-out analysis in the SSL setting. As shown in Section 5.2, AdaMatch outperforms FixMatch+ in SSL with DomainNet224, but it is not clear which modification (random logit interpolation or relative confidence threshold) more contributes to this improvement. If the relative confidence threshold contributes a lot, it somewhat matches the results in [R2].

- Table 1 provides a good summarization on the relationship between SSL, UDA, and SSDA, but there should be one important aspect that is not written here: which dataset can be limited in practice. In the setting of SSL, the number of labeled (source) data should be small as shown in the experiments. In contrast, in the setting of DA, the number of target data is often small, and this is the reason why some recent studies have been focusing on zero/few-shot domain adaptation. Since AdaMatch is based on FixMatch, it seems to implicitly assume the availability of abundant target data (e.g. uratio >> 1), which might be unsuitable for the DA setting. From this perspective, it would be worth discussing when AdaMatch performs well or do not.



**Summary Of The Paper:**

This paper presents a unified solution for unsupervised domain adaptation (UDA), semi-supervised learning (SSL), and semi-supervised domain adaptation (SSDA). The authors extend the state-of-the-art SSL method, that is FixMatch, to make it capable of handling DA setting where target data stem from a different distribution with source data. Experimental results show that the proposed method performs on par or significantly better than respective state-of-the-art methods in UDA, SSL, and SSDA.


**Summary Of The Review:**

Although the technical novelty is somewhat marginal, this study has a significant impact in the literature of SSL and UDA by empirically showing a unified scheme capable to solve both SSL and UDA. I vote for "weak accept."

---

> ### Author Response · Authors · 2021-11-12
> **Response to Reviewer aiq7**
>
> We thank the reviewer for their time and feedback on our paper. We respond to some of the reviewer's comments:
>
> 1. *Technical novelty:*
>     - Our contribution is that we introduce a single algorithm that achieves SOTA performance across all three domains of UDA, SSL, and SSDA. We show that the new techniques we introduce are able to help us achieve this goal. Our experiments are also helpful for the community in that they benchmark popular methods on all three domains so that future work can better understand which methods succeed where without having to run the 2,400+ experiments we performed.
>     - We do not dynamically change the threshold in the same way as [R2]. In our paper, the neural network itself influences the threshold rather than a user-defined schedule that slowly increases the confidence threshold throughout training.
>
> 2. *Relative confidence thresholding in early training:* While the reviewer makes a good point, the idea of most SSL techniques is that even if initially the model makes incorrect predictions, they'll converge to be good later on, and that's all we care about.
>
> 3. *Exclude-one-out for SSL:* The focus of this paper was to see how well SSL techniques could apply to problems with distribution shift. For this reason we spent most of our resources on UDA setting. While we agree with the reviewer that these experiments could be interesting in the SSL setting as well, we are limited in our experiments and so did not consider this configuration.
>
> 4. *Small amount of target domain data:* Our paper is focused on the setting where there is a reasonable amount of unlabeled data. If there is only a very small amount of target domain data, then some other methods might be necessary.

---

### Official Review · Reviewer_rNsu · 2021-11-04

**Correctness:** 4
**Technical Novelty And Significance:** 3
**Empirical Novelty And Significance:** Not applicable
**Recommendation:** 8
**Confidence:** 5

**Main Review:**

The advantages of this paper are as follows.
1) This paper proposes a neat algorithm that extends SSL to address the DA problems. The proposed method is technically sound, simple yet highly effective.
2) Extensive experiments are conducted to evaluate the proposed method by comparing representative existing ones.
3) The proposed method significantly extends the state-of-the-art performance.

I like this paper, but there are some minor issues that I hope the authors can clarify or improve in future versions.
1) In several places, the authors mentioned that early stopping is applied or the mean accuracy is calculated over many checkpoints. Does it imply that the performance of the proposed method is unstable and can vary significantly over training time?
2) Why not evaluate the method on the commonly used benchmarks? For both UDA and SSDA, there are more commonly used datasets and extensive prior work have reported performance on those datasets. How about the performance of the proposed method on those datasets? Is it still as competitive as in the datasets chosen in the paper?
3) Some techniques have not been ablation-studied, for example, the warmup function, applying cross-entropy twice, and prevention of gradient back propagation.


**Summary Of The Paper:**

This paper proposes to extend semi-supervised learning techniques to address domain adaptation problems and proposes a unified method, AdaMatch, that can handle unsupervised domain adaptation (UDA), semi-supervised domain adaptation (SSDA) and semi-supervised learning (SSL). While extensively borrowing techniques from SSL, AdaMatch proposes three unique techniques, i.e., distribution alignment,  random logit interpolation, and relative confidence threshold. Extensive experiments verify the efficacy of AdaMatch for the significant improvement over existing methods for different problems.

**Summary Of The Review:**

This is a good paper overall. It proposes a simple yet highly effective method to address various DA problems. There are some issues in the experiments, which I hope the authors could address in the response or in the future version. But overall I like it and recommend for acceptance.

---

> ### Author Response · Authors · 2021-11-12
> **Response to Reviewer rNsu**
>
> We thank the reviewer for their time and feedback on our paper. To address some of the reviewer questions:
>
> 1. We only apply early stopping when we use pre-training (where it is standard practice). Without pre-training, the performance does not vary significantly over the last several epochs and calculating mean accuracy over many checkpoints only serves to improves the reliability of our evaluations. Figure 5 in Appendix C illustrates typical performance of the model over the course of training in both the pre-training and standard setups).
>
> 2. We chose DigitFive since it is commonly evaluated in prior work. We use DomainNet because it is a larger, richer, more difficult task and we hope to encourage the field to evaluate more on DomainNet in the future. If the reviewer can point to specific datasets we should evaluate on we can try to include these results.
>
> 3.  - The warmup function is not very interesting, it's common practice and is mostly useful for initiating the convergence process, therefore we didn't include an ablation in the interest of space.
>     - Assuming you are referring to equation 9, then we are just applying cross entropy to the weak and strong augmentations. We could just write this as one cross entropy applied to a batch-concatenation, but thought it was mathematically cleaner to write it this way.
>     - If the reviewer means the stop_gradient in equation 10, this is common practice in all prior work dating back to the Pseudo Labels paper. It is already known in the literature that, without the stop_gradient, the pseudo label (which is supposed to be higher quality) will be affected by the logits (which is lower quality) and will therefore degrade.

---

### Decision · Program_Chairs · 2022-01-20

**Decision:**

Accept (Poster)

**Comment:**

Thanks for your submission to ICLR!

This paper presents a novel way to combine domain adaptation with semi-supervised learning.  The reviewers were, on the whole, quite happy with the paper.  On the positive side, the results are very extensive and impressive, it's a clever way to combine domain adaptation and semi-supervised learning, and it's a fairly general approach in that it works in several settings (e.g., unsupervised vs semi-supervised domain adaptation).  On the negative side, the approach itself is somewhat limited technically.

After discussion, the one somewhat negative reviewer agreed that the paper has sufficient merit and should be accepted; thus, everyone was ultimately in agreement.  I also read this paper carefully and personally find it very interesting and promising, so I am happy to recommend acceptance.  It seems to give state of the art performance in several cases, and could possibly lead to more research down the road on methods to combine adaptation techniques with SSL.